# Continuous, near real-time observations of water stable isotope ratios during rainfall and throughfall events

Barbara Herbstritt, Benjamin Gralher, Markus Weiler

Hydrology, Faculty of Environment and Natural Resources, Albert-Ludwigs-University, Freiburg, 79098, Germany

*Correspondence to*: Barbara Herbstritt (barbara.herbstritt@hydrology.uni-freiburg.de)

**Abstract.** The water isotopic composition of throughfall is affected by complex diffusive exchange with ambient water vapour, evaporative enrichment of heavy isotopes, and mixing processes in the tree canopy. All interception processes occur simultaneously in space and time generating a complex pattern of throughfall depth and water isotopic composition. This pattern ultimately cascades through the entire hydrologic system and is therefore crucial for isotope studies in catchment

hydrology where recharge areas are often forested while reference meteorological stations are generally in the open. For the quasi real-time observation of the water isotopic composition ($\delta^{18}$O and $\delta^2$H) of both gross precipitation and throughfall we developed an approach combining a membrane contactor (Membrana) with a laser-based Cavity Ring-Down Spectrometer (CRDS, Picarro), obtaining isotope readings every two seconds. A setup with two CRDS instruments in parallel analysing gross precipitation and throughfall simultaneously was used for the continuous observation of the temporal effect of

interception processes on the stable isotopes of water. All devices were kept small to minimize dead volume with time-lags of only four minutes for water from the rainfall collectors to the isotope analysers, to increase the temporal resolution of isotope observations.

Complementarily, meteorological variables were recorded at high temporal resolution at the same location. The achieved evolution from discrete liquid or event-based bulk samples to continuous measurements allows for direct comparison of

water stable isotope data with common meteorological measurements. Future improvements of the spatial representativeness will make our approach an even more powerful tool towards detailed insight in the dynamic processes contributing to interception during rainfall events.

## 1 Introduction

Stable isotopes of water ([18]O and [2]H) are ideal tracers due to the fact that they are part of the water molecule itself

(Gonfiantini, 1986). Thus, any change of the stable isotope ratios reflects the conditions a given water reservoir has been exposed to prior to sampling. This makes stable isotopes advantageous to e.g. solute concentrations, which may be prone to precipitation, degradation or chemical reactions. They have proven to be powerful tools for the characterisation of water flow paths and transport processes with a long record of applications at different spatial and temporal scales and in all parts of the water cycle. The isotopic composition of precipitation ultimately cascades through the entire hydrologic system, affecting

soil water, groundwater, evapotranspiration, and stream water isotopic signatures. Knowledge about the isotopic composition of precipitation is therefore crucial for isotope studies in catchment hydrology (Kendall and McDonnell, 1998; Vitvar et al., 2005). Stable isotopes have been used in numerous catchment studies for hydrograph separation (McDonnell et al., 1990; Uhlenbrook and Hoeg, 2003; Klaus and McDonnell, 2013; Fischer et al., 2017) and calculations of mean transit times and

travel time distribution (McGuire and McDonnell, 2006; Rinaldo et al., 2011; Heidbüchel et al., 2012; Kirchner, 2019). They were also widely used for end-member mixing analysis (Liu et al., 2004; Klaus and McDonnell, 2013). Recently, the application of water stable isotopes for calculating young water fractions has been described (von Freyberg et al., 2018; Stockinger et al., 2019; Kirchner, 2019). In subsurface hydrology, stable isotopes of water have been successfully applied to determine soil evaporation (Allison, 1982; Braud et al., 2009), groundwater recharge rates and sources (Blasch and Bryson,

2007; Adomako et al., 2010; Koeniger et al., 2016), They have also been used to identify flow paths (Uchida et al., 2006; Garvelmann et al., 2012; Stumpp and Hendry, 2012), mixing processes (Thomas et al., 2013), transit times (Stumpp et al., 2009; Timbe et al., 2014; Sprenger et al., 2016), root water uptake patterns (Rothfuss and Javaux, 2017), and hydraulic lift (Meunier et al., 2018).

Many studies have used the temporal dynamics in the isotopic composition of precipitation for estimating catchment

residence times. Particularly, on forested sites where meteorological and isotopic reference stations are generally in the open interception losses and accompanying isotope effects must be considered as they have a significant impact on the input function (Xu et al., 2014; Stockinger et al., 2015; Allen et al., 2017). The importance of understanding rainfall interception processes is presented in a thorough review by Allen et al. (2017). The reasons for the differences in depth and isotopic composition of gross precipitation ($P_g$) and throughfall (TF) are complex, since multiple interacting processes affect TF.

They are driven by evaporation from the canopy during or between storms as well as by diffusive exchange with ambient vapour. Furthermore, redistribution in the canopy and storage effects where water is differentially retained or mixed contribute to interception. Also sub-canopy water recycling i.e. evapotranspiration and re-condensation (Green et al., 2015) as well as mixing with water from previous events (Allen et al., 2014) has been described. These effects result in offsets of TF, they reduce the depth of TF compared to $P_g$ and cause the difference in $\delta^{18}O$ between event-based sampled TF and $P_g$.

Ignoring the differences between $P_g$ and TF depth and isotopic composition may flaw a range of isotope applications in forested catchments. In hydrograph separation, changes of runoff isotopic composition have to be evaluated in relation to the span between the endmembers baseflow isotopic composition and the water effectively recharging the catchment, which is the enriched TF in forested catchments. Ignoring the difference between $P_g$ and TF resulted in misestimation of "old water" of up to 10% (Kubota and Tsuboyama, 2003). The estimation of stream water transit time distributions (TTD) can also be

flawed by ignoring the difference between $P_g$ and TF. Stockinger et al. (2015) found generally lower travel times when using $\delta_{TF}$ instead of $\delta_{Pg}$ as model input. Additionally, they found more pronounced differences in TTDs when using $\delta^2H$ instead of $\delta^{18}O$ data. Similar effects can be expected for other isotope applications using precipitation data as model input or as substitute for the soil water isotopic composition. Due to interception-induced enrichment, the latter will be shifted towards higher values. This may affect evapotranspiration partitioning studies (e.g. Haverd et al, 2011; Rothfuss et al., 2012; Dubbert

et al., 2013, 2014). Also for the interpretation of the seasonal origin of soil water (Allen et al., 2019) it is crucial to use TF isotopic composition. Further, the percolation velocity of seasonal precipitation input fractions will be reduced as a consequence of the reduced precipitation input depth. This likely has an impact on plant water uptake and rooting depth studies using water stable isotopes. In this context, Goldsmith et al. (2018) also emphasized the relevance of throughfall

spatial input patterns. Based on rainfall depth data, canopy water storage capacity rather than evaporation has been found to be the main constituent of rainfall interception. A comparison of different methods for estimating the water storage capacity yielded significantly diverging results (Klaassen, 1998). This was concluded to have implications for a wide range of models describing soil-vegetation-atmospheric transfer. In ungauged catchments or when only $P_g$ instead of TF data are available, isotopic correction factors were determined empirically (Stockinger et al., 2015; Calderon and Uhlenbrook, 2016) serving as

surrogates to compensate for the lack of respective TF isotope data.

High spatial intra- and inter-storm variabilities have been found in depth and isotopic composition of TF. A synthesis study analysed the spatiotemporal variabilities of TF from 18 selected studies at a global scale. The study showed that the spatial patterns of TF were very heterogeneous and ecosystem dependent when related to leaf area index (LAI) or other biotic factors (Levia et al., 2011). These factors were also investigated by Brodersen et al. (2000). They found that depending on

the species (spruce and beech) and on the density of the vegetation cover the volume weighted mean of the relative interception loss was in a range of 12% to 41%. It was typically higher for small events and generally led to enrichment of heavy isotopes in TF. The spatial variability was investigated using a set of 94 TF collectors at three different forested sites, covering different types (coniferous and deciduous) and ages of trees, canopy densities and canopy diameters (Keim et al., 2005). There the authors investigated the spatial dependence of TF depth (storm-total) for three to seven storms in a six

20  months period with a geostatistical approach finding a high spatiotemporal persistence. The representation of high spatial variability in collected TF stable isotopic compositions could be improved by using a set of roving collectors (Allen et al., 2015). Although hypothesized by the authors, intra- and inter-storm variabilities of TF depth did not necessarily correspond with variations in isotopic composition. Consequently, collecting representative TF input data for isotopes studies is still missing and an observational challenge (Allen et al., 2015).

Traditionally, the isotopic composition of liquid water is determined with discrete samples being analysed in the laboratory, hence conducting isotope studies always implied a trade-off between limited spatio-temporal resolution and extensive (and expensive) lab work. With the recent development of laser-based isotope analysers like off-axis integrated cavity output spectroscopy (OA-ICOS) or Cavity Ring-Down Spectroscopy (CRDS), there is now the possibility to analyse water stable

30  isotopes faster and less expensive. The fact that water vapour is analysed directly 'as water' together with the field-deployability of the analysers and the virtually instant availability of isotope readings made way for several attempts aiming at in-situ isotope observations with high temporal resolutions. Berman et al. (2009) and Pangle et al. (2013) combined an autosampler with a flow-through system and were able to reveal otherwise unnoticed fine-scale (5-minutes) variations of precipitation isotopic compositions. A commercially available VALCO® valve unit coupled with laser spectrometers for

high-resolution sampling (9.5-minutes) was used by Leis et al. (2011) to investigate spring water isotope dynamics. Koehler and Wassenaar (2011) employed a marble-filled equilibrator and a minimodule device for producing and subsequently analysing a constant stream of vapour being in isotopic equilibrium with and therefore carrying in a known manner the isotopic information of the liquid phase of interest. Following the same principle, other researchers used gas-permeable

ePTFE surgical tubings for the investigation of precipitation trajectories (Munksgaard et al., 2012a) or seawater-freshwater mixing ratios (Munksgaard et al., 2012b). For the continuous investigation of rapid water isotope changes in a soil column experiment, Herbstritt et al. (2012) employed a commercially available hydrophobic membrane contactor for converting a small fraction of liquid water continuously into a stream of vapour, which was directly analysed by the coupled isotope analyser. However, none of these approaches have attempted to observe rainfall and throughfall in parallel.

Recent interception studies, mostly based on bulk sampling data and focusing on spatial variations, are especially lacking an appropriate temporal resolution to be comparable with the available temporal resolution of meteorological input data and hence to describe and better understand the physics controlling the differences in isotopic composition between $P_g$ and TF. Therefore, the aim of this study is to develop an approach for the analysis of $P_g$ and TF depth and isotopic composition at the point level at high temporal resolution based on the membrane contactor method, to compare and validate the continuous

isotope measurements with discrete liquid samples as well as with event-based bulk samples. With this approach the dynamics in depth and isotopic composition of $P_g$ and TF and hence, interception processes influencing depth and isotopic composition of TF can be investigated in unprecedented high temporal resolution.

## 2 Methods and Material

### 2.1 Sampling

We modified the setup developed and used previously for the in-situ observation of water stable isotopes in a soil column experiment (Herbstritt et al, 2012). A commercially available hydrophobic membrane contactor of 1 x 1 x 0.5 inch (MicroModule®, Membrana, Charlotte, NC, USA, www.liquicel.com) was combined with a CRDS isotope analyser (L2120-*i*, Picarro, Inc., Santa Clara, CA, USA, www.picarro.com). The contactor, originally designed for degassing liquids, was used in the so called 'sweep-mode' in order to continuously transform a small fraction of liquid water of interest with flowrates of

5-30 mL/min, according to manufacturer specifications, into a water vapour stream. Inside the contactor a microporous, hydrophobic, PP-based membrane (A = 100 cm²) divides the liquid from the gaseous phase. At the membrane's surface, dry carrier gas (e.g. $N_2$) mixes with vapour diffusing from the liquid phase through the pores across the membrane. Moist air then leaves the contactor at the gas outlet port (Fig. 1, right) which is directly connected to the CRDS. In the analyser-controlled stream of moist air (flow rate ~35 mL/min), readings of water vapour, $\delta^{18}O$ and $\delta^2H$ are given every two seconds.

The method requires a thorough determination of the temperature to account for the temperature-dependent isotope fractionation factors of the membrane as described in detail by Herbstritt et al. (2012).

Several modifications of the original setup were made for the quasi real-time observation of water stable isotopes in rainfall. A standard rainfall collector was enlarged from 200 to 1810 cm² with a PP-funnel to ensure sufficient water flow in case of low rainfall intensities and also to account for small-scale spatial variabilities in TF. To protect against clogging by litter fall, a metal mesh with 1 x 1 mm covered the funnel. At the bottom end, a smaller funnel with a volume of 3 mL was installed. From there, a stream of water was pumped to the membrane contactor with a peristaltic pump at a constant flowrate of 5 mL/min while at the same time water exceeding this flowrate was spilled and collected via an additional funnel into a sampling bottle. This overflow was volume-weighted, contributing to the event-based bulk sample.

All connections were made by gas-tight PFA tubings with an inner diameter of 1 mm. Easily replaceable glass fibre syringe filters (pore size 1-2µm) were installed in line to protect the membrane contactor from clogging. Removal of smaller particles or biofilms inside the contactor could be facilitated by back flushing with deionized water as needed or periodical rinsing every 2 to 4 weeks with weak acids, respectively. Temperature of the water in the tubing just before the contactor was stabilized and kept constant at 16°C using a peltier element (UEPT-KIT3) and a controller (UR3274U5, both obtained from uwe electronic, Wachendorf, Germany, www.uweelectronic.de) to avoid super-saturation and condensation of the vapour on the way to the isotope analyser operated at room temperature. Vapour isotope data were recorded as soon as temperature and thus vapour concentration at the membrane contactor was stable, which was usually the case within 5 to 10 minutes after the onset of precipitation sampling. All tubings were kept as short as possible, facilitating a time shift of no more than four minutes between precipitation and the respective readings displayed by the isotope analyser (Fig. 1). During rainfall events, discrete liquid samples were taken every five minutes at the liquid outlet port of the membrane module and analysed later in the laboratory. The overflow and the excess water downstream of the membrane module were collected and analysed separately. From these isotope results volume-weighted means were calculated to represent bulk sample values for each continuously measured event. Additional bulk-samples were physically collected and analysed for events when low intensities or the occurrence during night-time prevented continuous analysis. In close proximity (1.5 m) to the collector a tipping bucket (R3, Onset Rain Gauge) was installed. Rainfall was logged in 0.2 mm increments (HOBO UA-003-64 Pendant Event Data Logger, Onset, HOBO®, Bourne, MA, USA, www.onsetcomp.com) and data was aggregated to 1-minute intervals. Two of these setups were installed with 10 m horizontal distance from each other, sampling gross precipitation ($P_g$) and throughfall (TF) under a deciduous tree (*Acer campestre* L.) separately. In total, 28 bulk samples and nine continuously analysed events were obtained in August-September 2015 and throughout the vegetation period (May-September) of 2016. In any case, the measurements were carried out during the period when the leaves had reached their full sizes, in order to minimize the influence of the growing season. The meteorological variables air temperature ($T_a$) and relative humidity (RH) were recorded in 15 m distance to the tree with a CS215 sensor and logged with a CR1000 data logger (both available from Campbell Scientific, Inc., Logan, UT, USA, www.campbellsci.com) every minute. Additionally, the meteorological variables rainfall depth ($P_g$), air temperature ($T_a$), relative humidity (RH), air pressure, wind speed (v), and wind direction were available in 10 minute resolution from a climate station 250 m away.

Figure 1

## 2.2 Analyses

All isotope data are expressed in $\delta$-notation calculated with the following equation:

$\delta = \left(\frac{R_{sample}}{R_{VSMOW2}} - 1\right) * 1000‰$ (1)

where VSMOW2 is the Vienna Standard Mean Ocean Water and R is the isotope ratio ($^{18}O/^{16}O$ or $^{2}H/^{1}H$). Calibration of the samples was conducted using three in-house standards with distinct isotopic compositions, -16.65‰, -9.59‰, and 0.51‰ for $\delta^{18}O$, -125.05‰, -66.50‰, and -2.40‰ for $\delta^{2}H$, referenced to the international VSMOW-SLAP scale (Craig, 1961). They were pumped consecutively through the contactor after each rainfall event until a plateau in the isotope readings was reached

(~10 minutes) and treated similar to the continuously sampled precipitation water. Hence, potential long-term changes of the membranes e.g. build-up of biofilms or mechanical changes (small cracks, fissures) at the membrane did not have an effect on calibrated isotope data. For data noise reduction of the continuous measurements we calculated moving averages with an integration time of 90 s. All liquid water samples were analysed on a CRDS laser spectrometer (Picarro L2130-$i$) with a post-calibration accuracy of $\pm$ 0.05‰ for $\delta^{18}O$ and $\pm$ 0.35‰ for $\delta^{2}H$.

D-excess ($d$) was used to indicate the deviation from the global meteoric water line (GMWL) and likely non-equilibrium fractionation by evaporation (Gat, 1996):

$d = \delta^{2}H - 8 * \delta^{18}O$ (2)

The difference in isotope characteristics between TF and $P_g$ is indicated by the symbol $\Delta$:

$\Delta\delta^{2}H = \delta^{2}H_{TF} - \delta^{2}H_{Pg}$ (3)

$\Delta\delta^{18}O = \delta^{18}O_{TF} - \delta^{18}O_{Pg}$ (4)

$\Delta d = d_{TF} - d_{Pg}$ (5)

Relative interception loss ($Loss$) is the difference in depth between $P_g$ and TF

$Loss = \frac{P_g - TF}{P_g} * 100\%$ (6)

Vapour pressure deficit (VPD) is calculated to indicate potentially high or low evaporation with the following equation

(modified from Foken, 2008):

$VPD = 6.107 hPa * e^{\left(\frac{17.62*T_a}{243.12°C+T_a}\right)} * \left(1 - \frac{RH}{100\%}\right)$ (7)

where RH is relative humidity in % and $T_a$ is air temperature in °C.

## 3 Results

One example rainfall event observed in this study (Fig. 2) had a total depth of 6.8 mm and started with high rainfall intensities followed by more moderate intensities which lasted for roughly two hours. Continuous vapour derived stable isotope measurements, isotope ratios of discrete liquid and liquid bulk samples are shown for both isotope ratios investigated to illustrate the increase of temporal information during one single rain event when analysing in high temporal resolution. The moving average of calibrated vapour derived data ranged between -5.45‰ and -7.53‰ for $\delta^{18}O$ and -27.36‰ and -45.64‰ for $\delta^2H$ with respective mean precisions of $\pm$ 0.26‰ and $\pm$ 1.53‰. The isotopic composition of the event-based bulk sample of this event falls in that range with -5.68‰ for $\delta^{18}O$ and -32.67‰ for $\delta^2H$. Calibrated data of the liquid samples taken simultaneously were in a range of -5.34‰ to -7.43‰ in the case of $\delta^{18}O$ and -28.73‰ to -45.95‰ in the case of $\delta^2H$. Mean absolute deviations between the moving average of the continuously analysed vapour data and the discrete liquid samples were 0.11‰ and 1.35‰ with standard deviations of 0.096‰ and 0.81‰ for $\delta^{18}O$ and $\delta^2H$, respectively.

Figure 2

Analysing all 28 event-based bulk samples in this study, the variables rainfall depth ($P_g$), mean rainfall intensity, interception loss, $\delta^{18}O_{Pg}$, $\delta^{18}O_{TF}$, the difference in deuterium excess ($\Delta d$) and the isotopic difference $\Delta\delta^{18}O$ are plotted against each other to check for correlations between them in a scatterplot matrix (Fig. 3). A significant (p-value < 0.05) but moderate negative correlation between the logarithm of the mean rainfall intensity and the relative interception loss indicates that the highest interception losses were found during events with lowest rainfall intensities. The interception loss ranged predominantly between 30% and 50%. Also a weak negative correlation between interception loss and rainfall depth was found, as well as a weak positive correlation between the logarithm of the mean rainfall intensity and rainfall depth. The isotopic composition of $P_g$ ranged from -1.58‰ to -11.69‰ for $\delta^{18}O$ and was significantly correlated to the respective isotopic composition of TF which ranged from -0.88‰ to -10.15‰. Only non-significant correlations were found between the other investigated quantities. The explained variance by any of the considered variables alone was generally small.

Figure 3

In Figure 4 (a) the difference of the isotopic signature between TF and $P_g$ ($\Delta\delta^{18}O$) of 28 bulk samples is shown. It was calculated from flux-weighted means of $\delta^{18}O$ of TF and $P_g$. The data of the bulk samples were grouped in classes of 0.5‰ increments. The maximum difference in $\delta^{18}O$ values of 2 - 2.5‰ was observed only for two events, while for 23 of the 28 events $\Delta\delta^{18}O$ was 1.5‰ or less. In contrast, in the shorter periods of continuous sampling, $\Delta\delta^{18}O$ values up to 3.5‰ were found (Table 1 and Fig. 4 (b)). Each symbol in Figure 4 (b) indicates the mean difference in $\delta^{18}O$ (y-axis), whereas the length of the symbol indicates the duration of the continuous sampling (x-axis), including start and end time of the respective

events. Two clusters could be identified with one containing three single rainfall events on a previously dry canopy, i.e. after at least 6 hours without rainfall, and the other containing six events on already wet canopies with markedly higher $\Delta\delta^{18}$O values.

Table 1: Depth, mean intensity, and isotope values of continuously sampled $P_g$ and TF. Data of differences in $\delta^{18}$O ($\Delta\delta^{18}$O (‰)) are reported as mean ± SD. (* tipping bucket malfunctioned)

|  | | | Rain ($P_g$) | | Throughfall (TF) | | TF-$P_g$ differences | |
|---|---|---|---|---|---|---|---|---|
| **Event #** | Starting time | Duration | Depth | Intensity$_{mean}$ | Depth | Intensity$_{mean}$ | Loss | $\Delta\delta^{18}$O |
|  |  | (h:mm) | (mm) | (mm/h) | (mm) | (mm/h) | (%) | (‰$_{VSMOW}$) |
| **1** | 2015/09/22 20:32 | 0:25 | 2.0 | 4.80 | 1.20 | 2.88 | 40.0 | 3.45 ± 1.16 |
| **2** | 2015/09/22 21:28 | 0:32 | 2.6 | 4.88 | 1.60 | 3.00 | 38.5 | 2.49 ± 0.98 |
| **3** | 2016/08/02 17:50 | 0:59 | 3.0 | 3.00 | (6.20)* | (6.20) | (-106.7) | 0.95 ± 0.37 |
| **4** | 2016/08/04 19:50 | 0:42 | 1.8 | 2.57 | 0.80 | 1.14 | 55.6 | 2.14 ± 0.5 |
| **5** | 2016/08/04 20:54 | 0:22 | 0.6 | 1.64 | 0.40 | 1.09 | 33.3 | 2.68 ± 0.27 |
| **6** | 2016/08/04 21:34 | 0:24 | 0.8 | 2.00 | 0.40 | 1.00 | 50.0 | 2.31 ± 0.26 |
| **7** | 2016/08/04 22:10 | 0:15 | 0.4 | 1.60 | 0.20 | 0.80 | 50.0 | 3.31 ± 0.35 |
| **8** | 2016/08/05 18:15 | 0:08 | 0.6 | 4.50 | 0.40 | 3.00 | 33.3 | 0.86 ± 0.32 |
| **9** | 2016/08/20 18:37 | 0:46 | 4.0 | 4.80 | 2.80 | 3.36 | 30.0 | 1.00 ± 0.33 |

Figure 4

Time series of (a) $\Delta\delta^{18}$O, (b) $\Delta\delta^{2}$H, and (c) $\Delta d$ calculated from continuous isotope data of TF and $P_g$ of all nine continuously observed events listed in Table 1 are illustrated in Figure 5. Values during events on initially dry canopies did not exceed 1.5‰ and 10‰ for $\Delta\delta^{18}$O and $\Delta\delta^{2}$H, respectively, with $\Delta d$ values ranging from +1‰ to -7‰. In contrast, values from events on wet canopies were in the range of +1.5‰ to +5‰ for $\Delta\delta^{18}$O and +11‰ to +43‰ for $\Delta\delta^{2}$H with $\Delta d$ values ranging from +12‰ to -6‰.

Figure 5

The combination of collector funnel area (1810 cm²) and water flow rate (5 mL/min) resulted in a threshold rainfall intensity of 0.03 mm/min that was required to ensure an air bubble-free stream of water being pumped to the membrane contactor. Thus, periods of pure water flow at both ($P_g$ and TF) contactors alternated with periods when air bubbles appeared at either one or both of the contactors in this example event (Fig. 6). Since the presence of bubbles proved to flaw isotope readings,

calculations of $\Delta\delta$ $^{18}$O, $\Delta\delta$ $^2$H, and $\Delta d$ were only reasonable for periods without bubbles at any contactor. The variability of the continuous P$_g$ isotope data was higher than the variability of the continuous TF data, where the signal was more dampened over time. Relative to P$_g$, TF became increasingly enriched in heavy isotopes during the event. The variability of $d$ was in the range of about 5‰ for both P$_g$ and TF, fluctuating initially in a corridor between 10‰ and 15‰. However, $d_{Pg}$

showed a negative trend and decreased by about 10‰ over the course of the event whereas $d_{TF}$ did not follow this trend and remained rather constant. Air temperature as well as vapour pressure deficit decreased with event duration.

Figure 6

**4 Discussion**

**4.1 Continuous measurements**

The modified setup of the method developed by Herbstritt et al. (2012) adapted to continuous rainfall and throughfall isotope measurement worked quite well in terms of providing continuous, thermo-regulated flows of water to the membrane modules and delivering reliable liquid water stable isotope data. The latter became evident by the good agreement of

continuous measurements and discrete liquid samples (Fig. 2 and also Fig. 6) which was persistently in the order of the measurement uncertainty for both isotope ratios under investigation. Given the dead volume of the perfused components the collected isotope data were subject to averaging with a time window of ~36 s. Additionally, we calculated a moving average with 90 s integration time for data noise reduction. Nonetheless, large intra-storm variabilities exist in the isotopic signature of P$_g$ and TF. They would have been impossible to detect when solely relying on commonly taken event-based bulk samples

or even on data representing a higher sampling interval of typically 5 mm precipitation depth. For the sampling points in our study, we found that the variability of the continuous P$_g$ isotope data was higher than the variability of the continuous TF data (Fig. 6). For the latter, the signal was more dampened probably due to mixing processes in the canopy, which was also found in other studies (Qu et al., 2013). The lower intensity and total depth of TF as compared to P$_g$ indicates evaporation from the canopy. However, this is not reflected in continuous $d_{TF}$ values (Fig. 6). We would have expected that they follow

the trend of $d_{Pg}$ values while being comparatively lower as a result of non-equilibrium fractionation. Neither was the case.The dynamics in the isotopic composition in P$_g$ as well as in TF could be captured by the continuous measurements and also to some degree by the 5-minute discrete liquid samples. This provides a good evidence for the applicability of the developed continuous method. On the other hand, it is also an indication that a temporal resolution of around five minutes might be sufficient to capture the isotope dynamics of rainfall events, if continuous sampling is not possible. However,

continuous data were instantly available whereas data of sampled liquid water were available not before conventional analysis via CRDS was completed, which lasted for another two days. Additionally, continuous measurements dispense the

transport and storage of large quantities of samples. Generally, there was a tremendous loss of information about short term dynamics or trends when comparing the results with commonly taken bulk samples or 5 mm depth incremented samples.

## 4.2 Bulk sample data

In the data of the collected bulk samples the isotopic composition of $P_g$ and TF were highly correlated to each other. This
concurs with our expectation, as does the quasi persistent enrichment in heavy isotopes of TF relative to $P_g$ (positive $\Delta\delta^{18}O$ values) indicating evaporation from the canopy. Although expected, no significant positive correlation between $\Delta\delta^{18}O$ and relative interception loss was found. The same applies for potential negative correlations between $\Delta d$ and $\Delta\delta^{18}O$ or relative interception loss, which were also not found. However a moderate negative correlation between interception loss and the logarithm of rainfall intensity, a weak negative correlation between relative interception loss and depth of incident rainfall as
well as a weak positive correlation between the logarithm of the mean rainfall intensity and rainfall depth could be observed (Fig. 3). This means that the highest interception losses were found during events with either the lowest rainfall intensities or with the lowest depths and that higher rainfall intensities result in higher total rainfall depths. This is in line with results found in other studies (Dewalle and Swistock, 1994; Brodersen et al.; 2000, Keim et al.; 2005, Kato et al., 2013; Allen et al., 2017). Only non-significant correlations were found among the other investigated quantities. The generally small explained
variance among any of the considered variables illustrates the complexity of the processes contributing to interception loss and the transformation of $P_g$ isotope ratios when becoming TF.

## 4.3 Meteorological correlations

Depletion in heavy isotopes in open site rainfall was observed in the last ~30 minutes of the event presented in Figure 2. Several effects could cause such a pattern. The amount effect, reported for lower latitudes (Moore et al., 2014), can be ruled
out due to the fact that at the same time rainfall intensity was quite low compared to other periods of this particular rainfall event. Also a rainout effect cannot be attributed to our data as it is only detectable on a spatial scale considering the movement of air masses and rain clouds. Rather, the simultaneous decrease of air temperature indicates the passing of a weather front which appears to be the relevant explanation for the observed changes in precipitation isotope values.
Meteorological variables also did not provide a clear single evidence for the observed enrichment, indicating that multiple
variables affected the isotopic processes in the canopy. Typically, air temperature as well as vapour pressure deficit being the main driver of evaporation slightly decreased over the course of an event. However, evaporation as evidenced by the observed interception losses still occurred and altered the isotopic signal. At the same time mixing of antecedent water with new precipitation water occurred in the canopy. Therefore, it remains unclear to what extent the increase in difference between synchronous $P_g$ and TF isotope data must be attributed to evaporative enrichment or to changing mixing processes
following the variable rainfall intensities (Keim and Link, 2018). Horizontal water redistribution in the canopy via flow along branches may result in mixing of different water reservoirs. This may have an effect on the observed isotopic

composition as it also changes the unknown spatial pattern of precipitation water isotopes (Levia et al., 2011). In addition, the spatial scale and variability of mixing, certainly at the leaf level, but probably also among leafs as water drips from leaf to leaf, needs to be considered if we would like to decipher the isotope dynamics caused by water mixing and evaporation in the canopy of a tree.

## 4.4 Deuterium excess

In the derived $\Delta d$ values two clusters could be observed that were similar to but less distinct than the difference in $\delta^{18}O$ and $\delta^2H$ of the continuous measurements. The time series of $\Delta d$ from initially dry canopies were quite stable and predominantly negative. In contrast, we observed higher, mainly positive and more fluctuating $\Delta d$ values from initially wet canopies (Fig. 5 (c)). The fluctuations may be attributed to changing meteorological conditions affecting evaporation but also to the fact that $\Delta d$ values are quantities derived from four isotope measurements ($\delta^{18}O$ and $\delta^2H$ of both $P_g$ and TF) and therefore bear generally higher uncertainties due to error propagation. In the case of initially wet canopies, mixing with pre-event water which was inconsistently subjected to evaporative enrichment of heavy isotopes may have contributed to the observed higher $\Delta d$ variabilities. The positive $\Delta d$ values are not consistent with our expectation that evaporation would persistently lead to negative $\Delta d$ values and it stands out that highest $\Delta d$ values were calculated from continuous samples with highest evaporative enrichment in heavy isotopes of both $\delta^{18}O$ and $\delta^2H$ (Fig. 5). Regardless of the sampling method, we found positive $\Delta d$ values in both continuous (Fig. 5 and Fig. 6 (qualitatively)) and bulk sample data (Fig. 3). This indicates that the appearance of positive $\Delta d$ values is not a methodological artefact due to our way of continuous data interpretation but had to be physically based. Conceptually, positive $\Delta d$ values could have resulted from evaporation lines with slopes higher than that of the meteoric water line, causing TF isotope values to plot above the meteoric water line in dual isotope space. However, we are not aware of such a phenomenon. Positive $\Delta d$ values as a result of mixing processes, as suggested elsewhere (Allen et al., 2017), would still necessitate at least one substantial endmember with significantly higher than evaporation-only-caused $d$ values. The hypothetical formation of such endmembers remains unclear and was not indicated by the continuous $P_g$ data.

## 4.5 Technical aspects

Due to required minimum flowrates at the contactor (5 mL/min), the used setup with a collector area of 1810 cm² was limited to minimum rainfall intensities of 0.03 mm/min (1.8 mm/h). The fact that some of the successfully investigated events appear to have intensities below 1.8 mm/h (Tab. 1) may be due to TF heterogeneities that came into effect at the different locations of the tipping bucket and the continuous TF sampling. For sampled intensities below the identified threshold value, a larger or multiple collectors would be necessary, whereby larger collection funnels in combination with low rainfall intensities will increase the risk of evaporative enrichment from the funnel surface, leading to methodological artefacts that need to be avoided. These effects, however, are typical for most throughfall collectors as large and long surfaces are preferred. Therefore, dimensioning the collectors always is a trade-off and the maximum size is limited.

On the other hand, larger or multiple collectors would ensure a better representation of the spatial variability of $P_g$ and TF isotope data. This issue is critical for the thorough investigation of spatially heterogeneous processes affecting isotope input data at e.g. the catchment scale. A better representation of spatially distributed isotope data based on the proposed method can be achieved by installing a representative number of smaller collectors from where water is continuously aggregated and directed to one analyzer. This approach would consider the spatial heterogeneity similar to the roving collector approach (Allen et al., 2015) but at the same time it still does not allow for the quantification thereof. For the latter purpose, the entire setup described in this study could be multiplied in order to analyze individually the continuous water samples from the respective number of collectors considered necessary for covering the spatial heterogeneity. However, given the required number of suitable isotope analyzers this approach may not be feasible for many research groups. It should be noted that the definition of a rainfall event is not consistent between continuously measured and bulk sampled events. Bulk samples cover the entire time period of rainfall regardless of intensity at the point of observation and mere existence of rainfall at the complementary observation point ($P_g$ vs. TF). In contrast, continuously measured events are defined by sufficient simultaneous rainfall intensity at both points of observation due to our setups' properties. Therefore, natural rainfall events can only partially be captured by continuous synchronous observations. At the same time bulk samples represent the flux-weighted mean of all conditions constituting the respective event. The total number of observed events is lower in the case of the continuously measured events due to a number of events with rainfall intensities below the threshold for the continuous sampling method and due to several events during the night when the sampling setup was not operated.

## 4.6 Data processing

Comparing $\Delta\delta^{18}O$ data during continuously measured events with those derived from event-based bulk samples, the differences were larger during the shorter continuously measured periods (Fig. 4). Nonetheless, in the continuously analysed dataset two clusters could be distinguished representing rainfall on already wet and on initially dry canopies (Fig. 4 (b)). The fact that wet canopies lead to an even stronger enrichment in heavy isotopes can probably be attributed to partly evaporated and therefore isotopically enriched pre-event water that mixed with new rainfall water. This supports the interpretation that antecedent conditions had a clear impact on isotopic enrichment of TF which was also described in previous studies (Keim et al., 2005; Allen et al., 2014; Stockinger et al., 2015; Allen et al., 2017). Relative to $P_g$, TF became increasingly enriched in heavy isotopes (Fig. 6), which is similar to our observations of $\Delta\delta^{18}O$ in bulk samples on dry and wet canopies (Fig. 4 and Fig. 5). Highest $\Delta\delta^{18}O$ values were found in the cases of the continuous observations. One reason could be that the continuously analysed events are shorter and therefore potentially capturing extreme values while bulk samples represent flux-weighted mean isotopic signatures of the entire periods of rainfall and throughfall. On the other hand, we are aware that the calculation and interpretation of synchronous $\Delta\delta^{18}O$ and $\Delta\delta^2H$ data are disputable given assumable, yet unknown, time lags between $P_g$ and TF. Within each rainfall event, also past trends and variabilities of the $P_g$ isotopic compositions must be assumed to be reflected in instantaneous TF isotopic compositions, but are not considered with the proposed approach.

However, a quite common intra-event $P_g$ isotopic depletion trend (Fig. 2 and Fig. 6) combined with any positive time lag between $P_g$ and TF would result in higher synchronous $\Delta\delta^{18}O$ values compared to estimates derived from bulk sample data. For continuous measurements the differences in the isotopic signature were consistently calculated from isochronous $P_g$ and TF data although we are aware that water falling from the canopy (i.e. TF) is always a mixture of new rainfall (i.e.

isochronous $P_g$) and rainfall that has occurred at different points in time before. The time lags depend on canopy storage capacity and rainfall intensity (Allen, 2017) and therefore vary within each single event as well as between different storms.

### 4.7 Potential modelling approaches

For the quantitative description of processes affecting throughfall depth and isotopic composition, and thus the possible explanation for the discrepancies in $\Delta\delta^{18}O$ values as well as intra- and inter-storm variabilities, a mechanistic modeling

approach could be envisioned. Such an approach could build on the one developed by Keim et al. (2006) and aim at the quantification of the intensity-dependent canopy storage capacity and hence variable time lags between $P_g$ and TF. The canopy storage capacity has been shown to be a function of leaf area and rainfall intensity (Keim et al. 2006) and may also depend on wind speed or other meteorological properties. While being attached to the canopy, water is exposed to meteorology-dependent, thus variable, evaporation. This caused the effective cumulative $P_g$ depth to decrease thereby

producing variable time lags or rather transit time distributions between $P_g$ and TF. Evaporation also causes enrichment in heavy isotopes of canopy-stored water (Fig. 3 and Fig. 4). Consequently, for every envisioned simulation time step the TF isotopic composition needs to be calculated consecutively taking into account the respective fractions of $P_g$ remaining from simultaneous and prior time steps and constituting the instantaneous reservoir releasing TF. In addition, internal mixing will occur at the leaf scale or inter-leaf scale. These mixing assumptions are probably least known and would need to be tested

under different conditions. Finally, all relevant model parameters can be assumed to be dependent on meteorological variables thus further complicating this modeling and emphasizing the importance of continuous precipitation data representing different climates and vegetation characteristics. Such a model would derive from $P_g$ data a more realistic isotope input function of water effectively recharging forested catchments. For validation purposes, it would then require high resolution meteorological and isotope data as available from our setup in order to match the resolution of the envisioned

modeling time steps.

### 5 Conclusions

We could demonstrate that the proposed method is suitable for continuously observing the water stable isotope dynamics in precipitation and throughfall. We facilitated a huge increase in temporal resolution compared to isotope assays based on bulk sampling. Our approach supersedes taking, transporting and storing liquid samples and at the same time provides data much

faster. The instant data availability enables immediate reactions during rainfall events while the operator is still in the field.

Employing our setup, the temporal resolution of the isotope data corresponds with the temporal resolutions that are already common in high frequency meteorological observations.

All components employed in this study are commercially available and can be installed with reasonable effort. In the present design the setup cannot yet be left unattended due to the necessity of periodical cleaning and maintenance like changes of the in-line filters. However, proper precautions excluding clogging by e.g. leaf debris should solve this issue as well. Due to the selected dimensions of our setup and the resulting minimum rainfall intensity of 0.03 mm/min, the system was not able to capture events with low rainfall intensities, e.g. most stratiform rainfall events, but this can be changed to make the approach suitable for a wider range of rainfall intensities.

Significant correlations of $P_g$ and TF depths and depth-derived quantities were found as expected and concurred with findings from other studies. The lack of significant correlations involving isotope derived quantities cannot be explained with our current knowledge and process understanding and calls for further scrutiny. Especially the high abundance of positive $\Delta d$ values should be subject of future studies. Additionally, knowledge about intra-canopy mixing and the time lags between $P_g$ and TF as required for a precise, physically based calculation of the evaporative enrichment still remains a challenge for future applications. We are therefore confident that our setup, especially when employed across larger spatial scales, will contribute to the aim of thorough isotopic sampling of TF. The obtained data will be crucial for mechanistic modelling approaches which will yield more realistic isotope input functions and thereby improve water flow and solute transport estimations for vegetated catchments.

## Author contribution

Barbara Herbstritt and Markus Weiler designed the research. Barbara Herbstritt investigated, performed and tested the technical implementation of the design. Benjamin Gralher contributed by advice. Barbara Herbstritt collected the data, evaluated the results and wrote the manuscript. Benjamin Gralher and Markus Weiler contributed by advice in evaluating the data and by reviewing the manuscript.

## Acknowledgements

We gratefully thank our technician Emil Blattmann for technical assistance. The article processing charge was funded by the German Research Foundation (DFG) and the University of Freiburg in the funding programme Open Access Publishing. We thank three anonymous reviewers for their constructive comments which helped to substantially improve the manuscript.

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

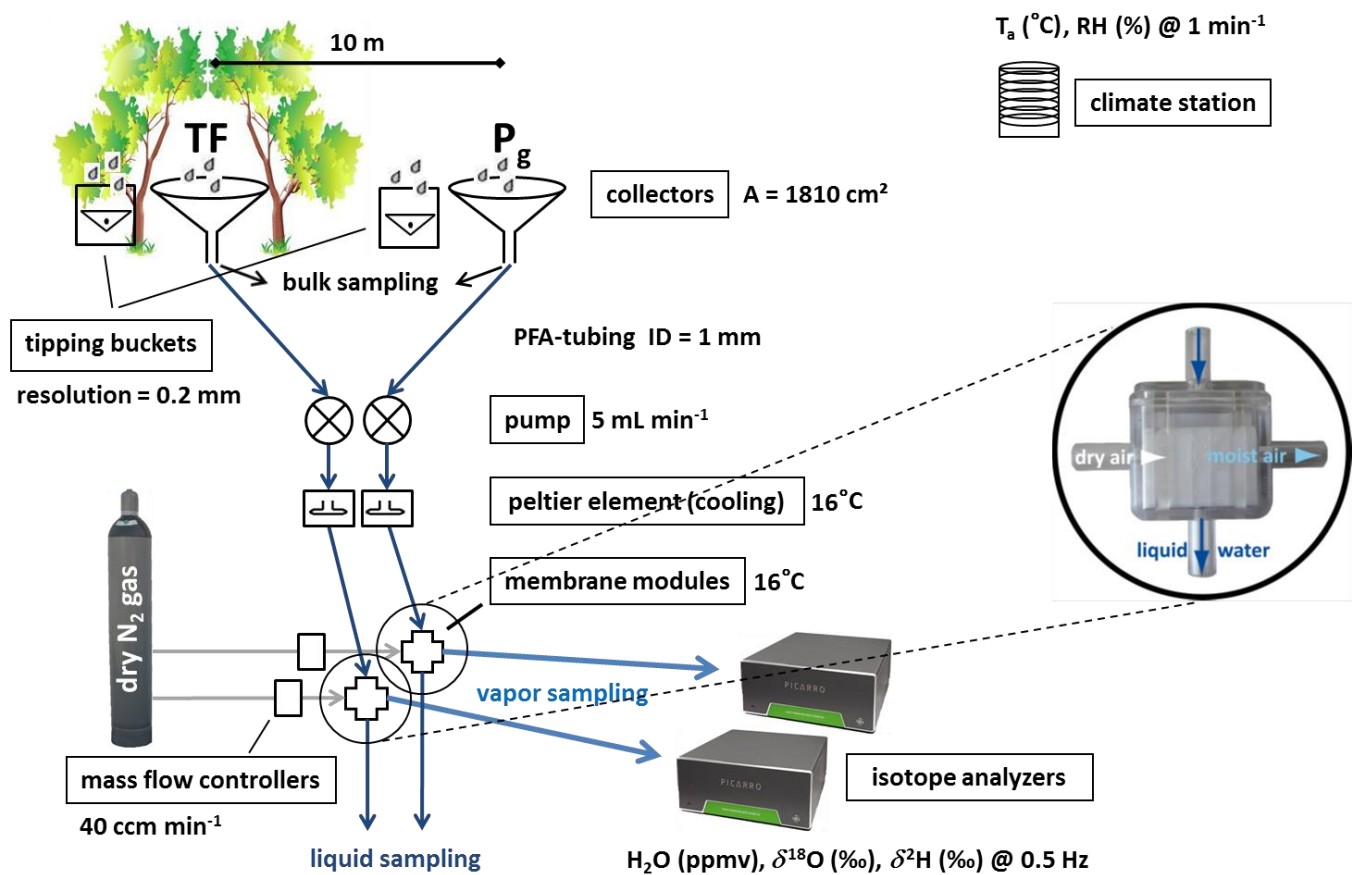

**Figure 1:** Setup for continuous water vapour stable isotope measurements of gross precipitation ($P_g$) and throughfall (TF) with two analyzers in parallel; membrane contactor (center-right) employed for continuous production of water vapour; discrete liquid and liquid bulk sampling of $P_g$ and TF; recording of rainfall depth (tipping buckets, upper left) and the meteorological variables $T_a$ and RH (upper right) every minute.

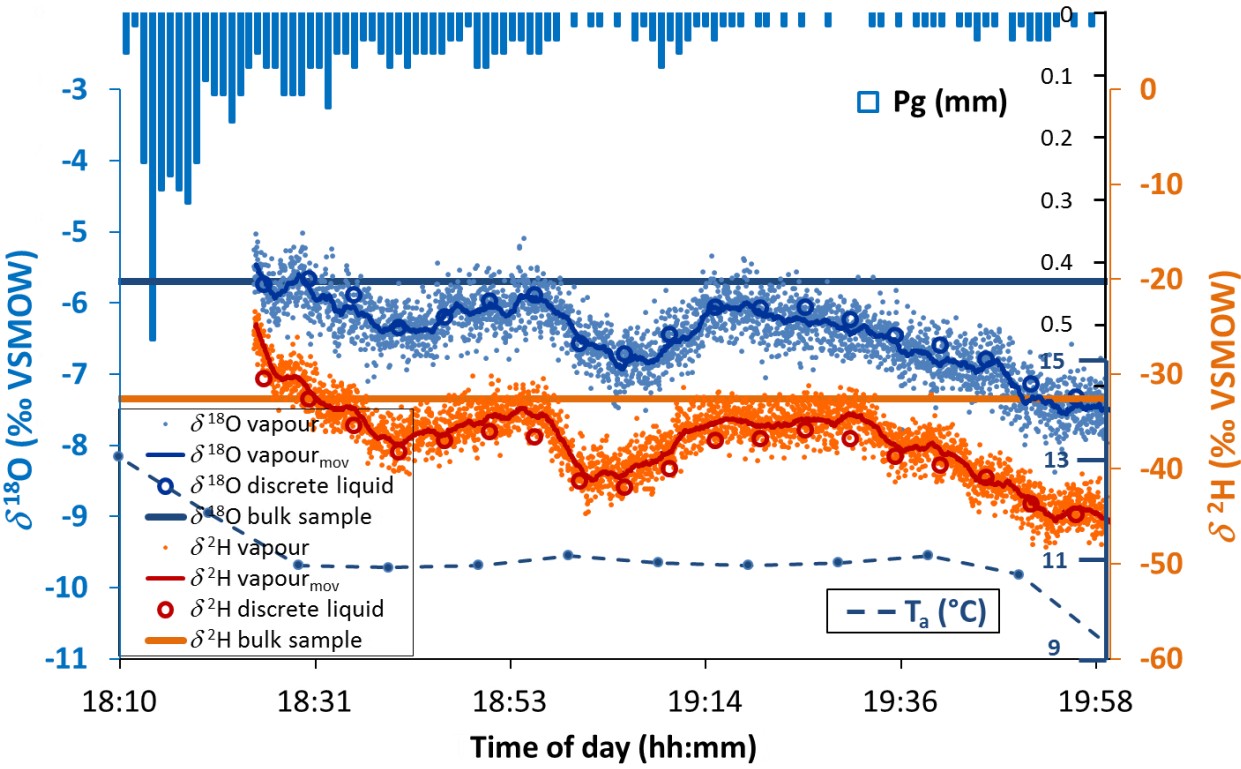

**Figure 2: Time series of rainfall (P$_g$) depth per minute (vertical bars); 10-minute data of air temperature in 250 m distance (T$_a$) (dashed line) during the rainfall event from April 17$^{th}$ 2015; data of $\delta^{18}$O in blue and $\delta^2$H in orange; vapour-derived data recorded every two seconds (small dots), starting after temperature at the contactor was stable, 90 sec. moving average (vapour$_{mov}$) (solid lines), discrete liquid samples (open circles) taken every five minutes, event-based bulk sample (horizontal bars).**

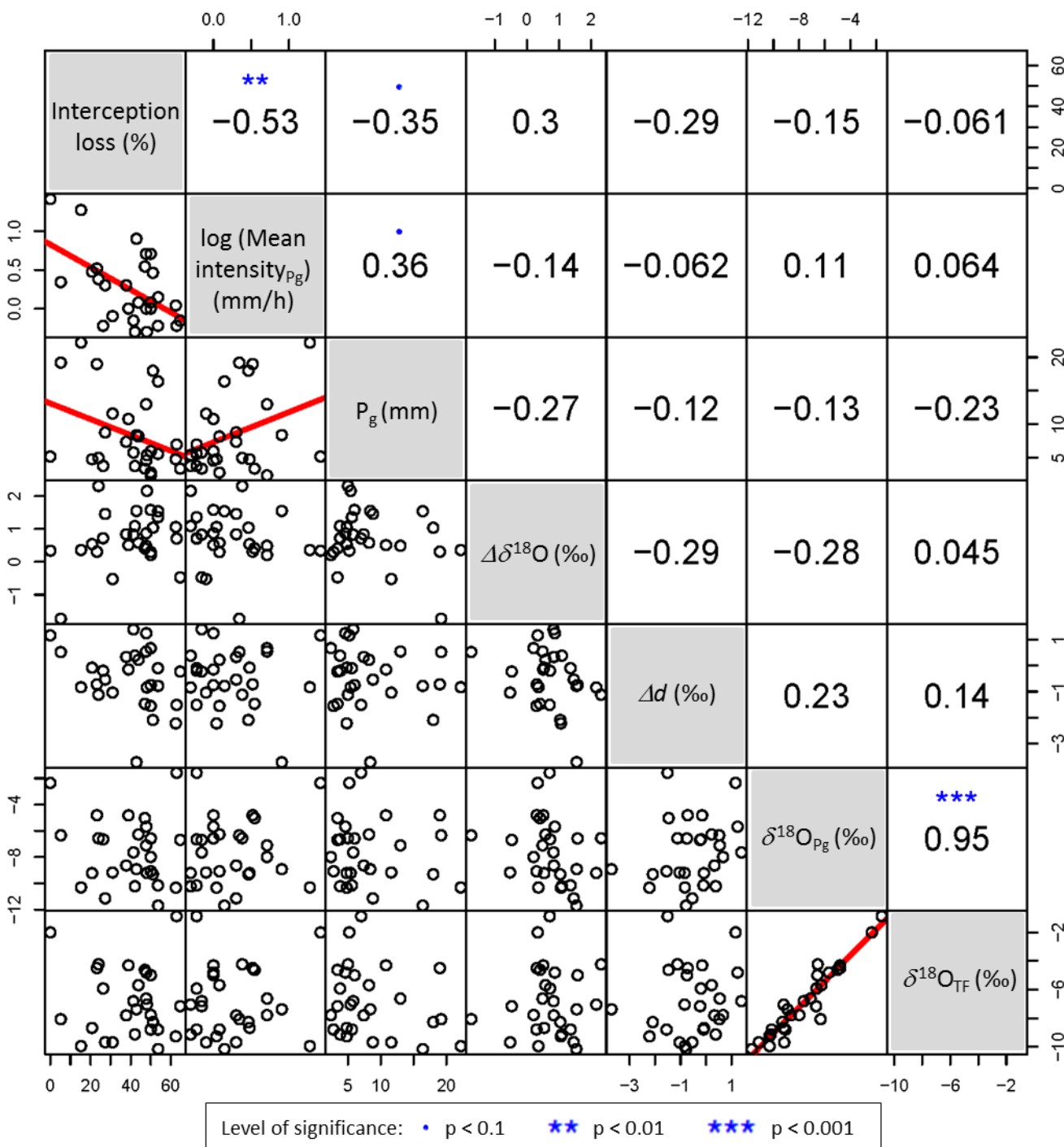

**Figure 3: Scatter plot matrix of relative interception loss, log of mean $P_g$ intensity, $P_g$ depths and isotope-related characteristics ($\Delta\delta^{18}O$, $\Delta d$, $\delta^{18}O_{Pg}$ and $\delta^{18}O_{TF}$) derived from 28 event-based bulk samples. Upper right part: Pearson correlation coefficients and level of significance (stars); lower left part: scatter plots and linear regressions (red lines).**

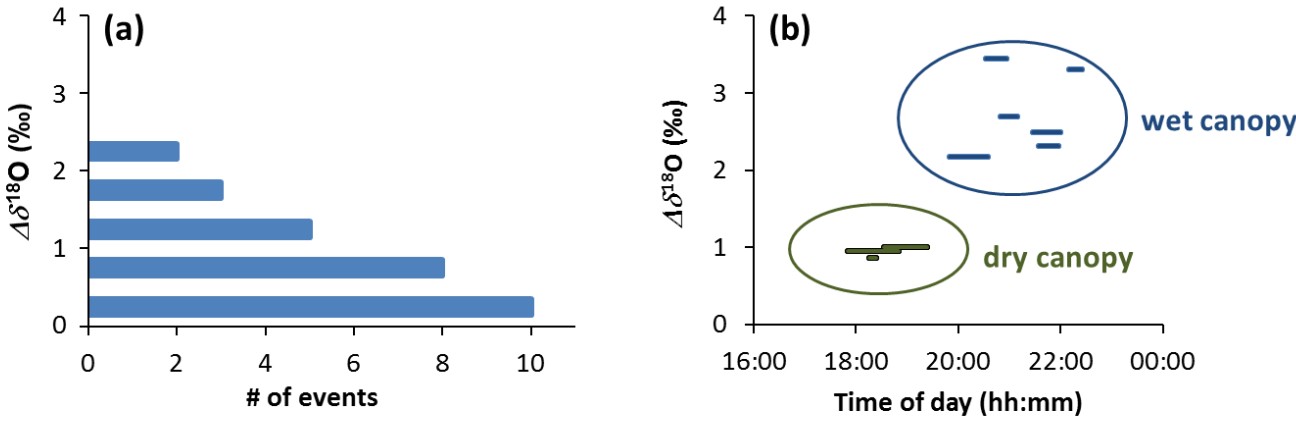

**Figure 4: Difference of the isotopic signature ($\Delta\delta^{18}$O) between TF and P$_g$ for (a) 28 event-based bulk samples and (b) 9 continuously analysed events of the same period.**

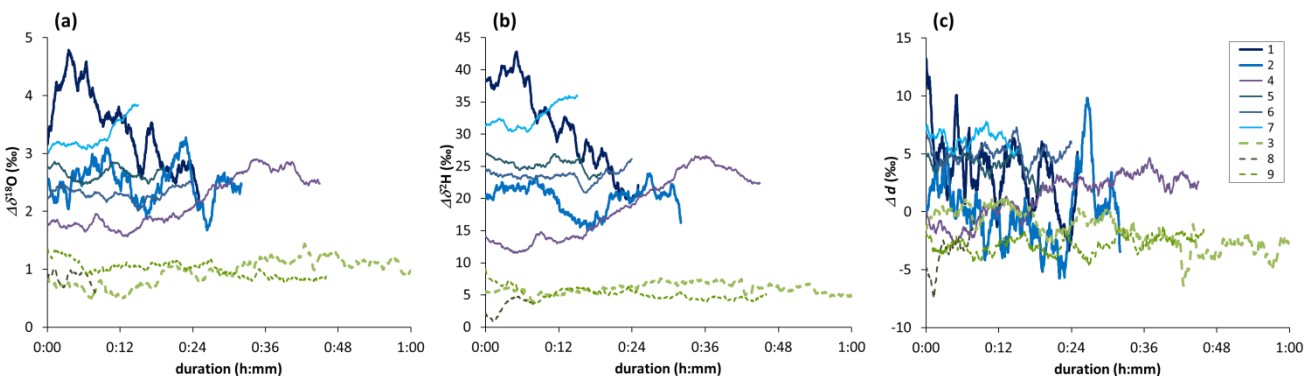

Figure 5: Time series of differences between the isotopic signature of TF and P$_g$, (a) $\Delta\delta^{18}$O, (b) $\Delta\delta^2$H, (c) $\Delta d$; events on initially dry canopy (dashed line) and on wet canopy (solid lines).

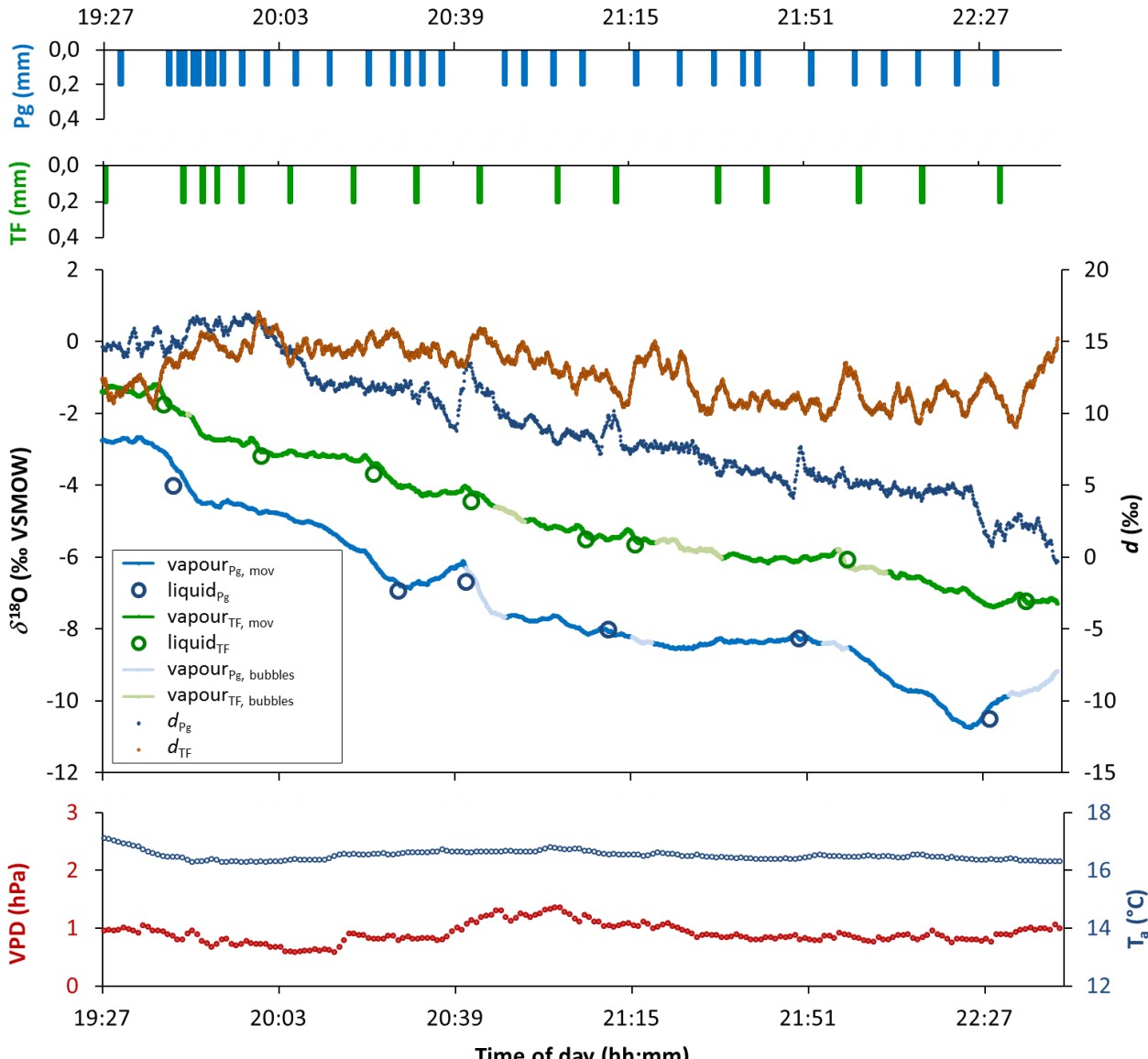

**Figure 6: Time series of liquid water $\delta^{18}O$ and d-excess ($d$) derived from vapour sampling of $P_g$ (blue) and TF (green), discrete liquid samples (big circles), air temperature ($T_a$) (small blue circles) and vapour pressure deficit (VPD) (red) of the rainfall event from August 4$^{th}$ 2016. Time series of $d_{Pg}$ (dark blue dots), $d_{TF}$ (brown dots) and $T_a$ are referenced on the right vertical axes. Periods of intensities below threshold for the continuous sampling method (bubbles at membrane contactor) are shown in light blue ($P_g$) and light green (TF).**