# Peer review of "Continuous, near real-time observations of water stable isotope ratios during rainfall and throughfall events"

_Hydrology and Earth System Sciences, 2018_

## Referee Comment (RC1) · Anonymous Referee #1 · 19 Jul 2018

In this manuscript, the authors present results of a novel method to determine stable isotope ratios of H and O (delta2H and delta18O values) in water of incident rainfall and throughfall below a selected individual tree in high temporal resolution making use of the latest developments in infrared laser spectroscopy.

Overall, the conducted research is sound and the manuscript is well structured. However, the language is sloppy and imprecise and needs to be considerably improved.

In the following, I offer a number of line-by-line comments to improve the manuscript before it can be accepted for publication in Hydrology and Earth System Sciences:

p. 1, l. 1-2: The title is misleading and incomplete. The measurement is not real-time but highly resolved (but with a temporal delay), the considered stable isotope ratios

must be specified, because rainfall and throughfall do not only consist of water but include numerous solutes, which also have to a large part several stable isotopes. Therefore, the title should be something like "Temporally highly resolved measurement of stable hydrogen and oxygen isotope ratios in water of rainfall and througfall with a novel infrared laser-based method" p. 1, l. 6: Like in the title the considered isotopes and their molecule need to be mentioned. p. 1, l. 6: It is unclear what the difference between exchange and mixing is. Add an explanation. p. 1, l. 8 (and throughout the manuscript): It is unclear what you mean by "amount". Do you talk about the rainfall/throughfall rate or volume? Be clear. p. 1, l. 10: What do you mean by "gross precipitation"? The water of the incident rainfall? p. 1, l. 16-18: This three-line statement has (almost) no content. Specify, what exactly makes your method to a tool for more insight. p. 1, l. 20: Tracers of what? Water sources? Water flow paths? Mixing processes? All of them? But then it would no longer be ideal, because usually a tracer is expected to be specific for a single source or a single process. p. 1, l. 21-22: This is again too unspecific. What exactly were water isotopes used for? p. 1, l. 24: What is crucial? The knowledge of the isotope ratios? But there was also a catchment hydrology before the isotopes could be measured and there are catchment hydrologists who do not use water isotope ratios. p. 1, l. 27: "important role" for what? Location, rate and volume of water input into the soil? p. 2, l. 1: The parentheses should be around the year only. p. 2, l. 3: Why should "redistribution in the canopy" have an isotope effect? p. 2, l. 5: The mixing was already mentioned in l. 4. p. 2, l. 6: Perhaps better "deciduous and coniferous" in context with "type", because spruce and beech are plant species. p. 2, l. 7: To what do the numbers refer? Interception loss? Throughfall in % of the rainfall? Vegetation cover? p. 2, l. 8: Enrichment of which isotopes in which compound? p. 2, l. 8-10: Is this really done? The following sentence is much more plausible. p. 2, l. 13-15: This sentence is confusing. I do not understand it. p. 2, l. 16-18: The temporal persistence of spatial throughfall patterns clearly depends on the length of the observation period and on the vegetation type. Please specify both. Furthermore, it is unclear what the cited authors

studied: throughfall volume, rate or water isotope composition? p. 2, l. 20: Are you talking about the same study that you cited just before? Who "hypothesized"? You? p. 2, l. 21: The collection of representative throughfall volume/rate data is a classic in ecosystem sciences and it is rather well known how it can be reached. See e.g., Kimmins, J.P., 1973. Some statistical aspects of sampling throughfall precipitation in nutrient cycling studies in British Columbian coastal forests. Ecology 54, 1008–1019. Do you refer to representative stable water isotope ratios? p. 3, l. 12-14: However, if you want to investigate the mean water isotope ratios of throughfall, you also need a high spatial resolution of your samples to collect a representative throughfall sample as stated earlier in your introduction. Just one pair of samplers above and below the canopy is not sufficient for this purpose. p. 3, l. 24-25: Perhaps, the explanation can be a little bit expanded to avoid that the reader has to consult the cited paper. p. 30: The spatial variability of throughfall cannot be appropriately measured with a single collector of limited size. Usually, a large number of collectors is needed to collect a representative sample. I suggest that you make clear that your results refer to a point measurement, which is very likely not representative for the throughfall at larger scale. p. 4, l. 19: Where was incident rainfall measured? Above the canopy or in an adjacent forest clearing? p. 4, l. 20: Acer campestre (in italics) L. - i.e. spell genus name with upper scale A and include author Abbreviation. p. 5, l. 10: What does this deviation from the meteoric line tell us? What is the reason for non-equilibrium fractionation? The d value only appears in Fig. 6 but is not addressed in the discussion. p. 5, l. 14 (and troughout the manuscript): Capdelta values are given without the lower case delta (i.e. DELTA18O, not DELTAdelta18O). p. 5, l. 20: Why did you chose exactly this event? Add properties of the event (date, total volume, intensity). p. 5, l. 21: "grab" samples, anyway being sloppy jargon, sounds strange in the context of water. I at least cannot grab water... p. 6, l. 13: Usually, one starts with the left figure and then goes to the right one. p. 6, l. 29: "cm$^3$" is not the unit of an area. p. 6, l. 33: "without bubbles at any contactor" p. 7, l. 11-13: This statement seems almost trivial. What do we really gain by this higher temporal resolution? How

do you interpret Fig. 2? I am a bit disappointed of the depth of the discussion. p. 7, l. 21-22 and Fig. 3: Why do you highlight a non-significant correlation between interception loss and ïĄĎ18O? Furthermore, you should not show a regression line for non-significant correlations. p. 7, l. 22: It is unclear what you mean by "There is no clear pattern..." Perhaps: "The explained variance by any of the considered variables alone was generally small, illustrating the complexity of the processes..." p. 7, l. 24-25: This sentence is confusing. It is clear that a bulk sample represents a mean isotopic signature, while at higher resolution the extreme values can be seen, which is trivial. p. 8, l. 1-2: Furthermore, there is a pronounced spatial variation of throughfall quantity and quality, which cannot be captured with a single collector (see above). p. 8, l. 4: This repeats l. 22 (as the whole Fig. 6 is repetitive of Fig. 2 – albeit for another arbitrarily selected rainfall event). p. 8, l. 7: extent p. 8, l. 11: The ultimate objective would be the prediction of the mean input signal of throughfall into soil for a whole forested catchment or even the spatial distribution of this input signal, which would require much more extensive instrumentation. p. 8, l. 14: I think that "point" level is more appropriate, because you cannot extrapolate your measurement with a single collector to a larger area. Plots I think of are at least 10 x 10 m large and on such plots you might easily need > 10 collectors to measure the mean throughfall properties with an acceptable uncertainty. p. 8, l. 18: I would really be keen to learn about what we can gain by measuring the stable isotope ratios of water at this high resolution. Can we distinguish different processes or even quantify the contributions of these processes to the total throughfall? p. 8, l. 22: I agree that this is crucial, but again point at the problem of spatial representativity of the measurements which cannot be reached with the approach used in this paper. p. 8, l. 28: I wonder whether the authors are aware of the modelling efforts of Rutter et al. (1971), Agric. Meteorol., Gash and Morton (1978), J. Hydrol. and Gash (1979), Q. J. R. Meteorol. Soc. I think that it could be a way forward to add an isotope module to these models. Fig. 3: Add how many events were sampled to the figure legend. Furthermore, part of the lettering is too small. Fig. 4: Number subfigures. Fig. 6: I suggest to combine this figure with Fig. 2. Both figures show stable

isotope results for arbitrarily chosen individual events but only Fig. 6 is accompanied by the necessary information about the (micro-)meteorologic conditions. Furthermore, the d values are only shown but not interpreted. Either you add an interpretation of these results or remove the d values entirely. Furthermore, I am confused by the legend stating "in vapour". I understood that you indeed measured isotope ratios in vapour produced from a liquid sample in your contactor but you referred these values back to the liquid sample via a temperature-dependent calibration function. Do you indeed want to show the isotope ratios in vapour (not referred back to the liquid sample)? Why?

---

## Referee Comment (RC2) · Anonymous Referee #2 · 20 Jul 2018

The manuscript "Real-time observations of stable isotope dynamics during rainfall and throughfall events" by Herbstritt et al. presents and discusses an experimental setup designed to monitor in parallel the stable isotope composition of rainfall (Pg) and throughfall (TF) at high resolution during several summer rainfall events. High resolution stable isotopes in rainfall have already been observed and documented in previous studies. Yet, this study is the first I am aware of that compared the rapid dynamics of rainfall and throughfall and looked into their mass weighted average difference over several rainfall events. In the abstract and the introduction, the authors summarize the importance, extent, and main reasons of the difference between isotopic composition of throughfall and that of rainfall. The authors justify the need for a comparison of these signatures at a higher resolution than in the past, which reflects their measurement

setup. This is described in the method section and well-illustrated in Figure 1. The results show time series of stable isotopes in TF and Pg, differences between them (dynamics and means), and an attempt to find correlations between such differences and meteorological variables. The discussion essentially deals with some of the limitations of the approach. The manuscript is clear and concise, and the figures are of good quality. Yet, no or only minor mechanistic understanding is provided. It will be a good contribution to isotope hydrology, after having addressed several comments and after some questions are clarified. The introduction and the discussion need to state clearer in what way this measurement setup can provide a more accurate estimate of the isotopic recharge in the catchments for typical applications in isotope hydrology. Why is it not enough to just consider the average mass weighted difference between isotopes in TF and Pg? What is a necessary detail of measurement? One figure that would improve in the manuscript in that regard is the relationship between precipitation intensity and $\Delta\delta$ for each measured storm. Are isotopic differences larger for higher intensities during a single event? A more detailed description of the applications of tracers in isotope hydrology is also needed in the introduction. The discussion needs to argue for which application this time-varying difference really is important. For instance, are End Member Mixing Analysis (Hooper et al., 1990), isotope hydrograph separation (Klaus & McDonnell, 2013), or travel time modeling (McGuire & McDonnell, 2006; Rinaldo et al., 2015; Hrachowitz et al., 2016) able to incorporate such high-frequency data and distinguish between Pg and TF? Some clarifications of details in the method sections will be necessary (see details below), as some important information was skipped. Furthermore, the English is somewhat bumpy especially in the first half of the introduction, and should be carefully revised before the manuscript is resubmitted. Eventually, one may consider to change the manuscript into a technical note, since many of the hydrological aspects are discussed rather briefly and the key contribution is the measurement of high-frequency variations in stable isotopes of Pg and TF and their characteristics. No mechanistic understanding is provided by the manuscript. The main conclusion also starts with the technological aspect.

Specific comments: Abstract line 15 The 4 min time-lag information can be confusing with respect to the 2 sec reading interval. Please make it clearer that the time-lag is the transfer time from the collector to the laser in the instrument. How does dispersion/diffusion potentially influence this? page 1: line 1ff. Please tighten up and do a better job in pointing out the relevance of isotopes from here until page 2 line 12. line 20 I would omit the word "signature", since the stable isotopes are the tracers, while their signatures are measurements. line 25 Move the citations (Kendall and McDonnell [. . .]) to line 22 after "water cycle" line 25 "There is. . . hydrology" I would omit that sentence which looks somehow too isolated, see previous comment. line 26 "residence times" is vague. Is it canopy, soil, or catchment residence times? Please be more precise. Line 28 New sentence after "Allen et al. . .". Delete "Since" Line 28 Citations after "forested" needed.

page 2: line 1 must be . . ."Allen et al. (2017)." line 13 delete "Typically"; found for what? Precip? TF? Runoff? References needed line 15 "spatial variability in general" is to general. Please elaborate line 17 Ref needed after "diameters". Replace "They" by "The authors" or "Keim et al." page 3: line 18 in-situ line 24,27 SPACE between numeric value and unit needed

page 4: lines 2-4 What is the dead volume inside smaller funnel just before the pump? How is it made sure that all the water exceeding the pump flowrate Qp is spilled into the bulk sample? In my perception, if Vd is the dead volume of the smaller funnel (let's assume Vd = 3 mL), then assuming complete mixing, the isotope signature effectively recorded is a moving average of the precipitation, with a time window of length Vd/Qp, i.e. about 36 sec. Please elaborate on this! lines 14-16 Were these discrete samples analyzed later in the lab? line 19 Is 10 m really sufficient to make sure that there are no effects of the trees at all on the gross precipitation? Did you see an effect of wind direction etc? line 20 How many events were recorded in total? It is never mentioned in the text. It also makes it difficult to follow the results. Add also more details about the events in tables.

[Figure]

page 5: lines 2-9 How was calibration applied? Did you apply an individual correction for each rainfall event based on the 3 measured standards? More details are needed here. Also, how did you ensure that there were no memory effects between standards when measuring them consecutively? line 6 What is meant by "long term changes in the membranes"? Please elaborate line 17 It should be mentioned here already why the VPD is calculated. line 20 What date did the event happened? This also needs to appear in the caption of Figure 2. Why not show directly the comparison between isotopes in Pg and TF in Figure 1, as in Figure 6?

page 6: line 1 It looks like the isotopes in Pg are getting lighter while rainfall intensities are getting lower. Is that not contradictory with the amount effect? lines 3-4 I suppose the interception loss is (Pg-TF)/Pg*100. It should be stated clearly how you calculated it. line 13 Were the interception losses and the $\Delta\delta\hat{}18$ O greater with time and plant growth from May to September? A plot with $\Delta\delta\hat{}18$ O in time during the growing season could be useful here. Any data on LAI? line 15 Is this mean difference flux weighted? I think this is important to emphasize. Line 21 "all events", see above, more information needed line 29: cm3 should be cm2

page 7: lines 1-2 Why is the TF signature more damped than the Pg signature? lines 2-4 Maybe it is because of the scale, but it does not seem like the VPD is decreasing on figure 6. Please clarify. Also, why not look at the relationship between VPD, Ta, and time-variable $\Delta\delta$ for all events? Some meaningful correlation could exist. lines 10-11 Some statistics about the differences between continuous measurements and the single liquid samples would be nice here to emphasize that point, even though it looks valid just when looking at the figures. For example, what was the average difference between the single liquid samples and the corresponding moving average values for each event? For all events? Does that vary a lot between events? lines 17-18 How are "wet", and "dry" canopies defined? line 25 So, why are the average differences in bulk samples and continuous samples so different? I think this is a crucial part of the manuscript showing why the measurement protocol proposed here is valuable! line 27

What process could explain that a wet canopy leads to an even stronger enrichment? Figure 1 Is the beginning of the event missed because of the stabilization of T? That info would be nice in the figure caption. Figure 5 A legend with the date of each event and the corresponding lines would be nice here. Figure 6 The points for d_Pg and d_TF in the legend are too small and hard to distinguish. The date of the event is missing.

Thanks for the interesting contribution to isotope hydrology!

References: Hooper, R.P., Christophersen, N., Peters, N.E., 1990. Modelling streamwater chemistry as a mixture of soilwater end-members – an application to the Panola Mountain catchment, Georgia, USA. J. Hydrol. 116 (1), 321–343. Hrachowitz, M., Benettin, P., van Breukelen, B. M., Fovet, O., Howden, N. J. K., Ruiz, L., et al. (2016). Transit times—The link between hydrology and water quality at the catchment scale. Wiley Interdisciplinary Reviews: Water, 3(5), 629–657. https://doi.org/10.1002/wat2.1155 Klaus, J., McDonnell, J.J., 2013. Hydrograph separation using stable isotopes: Review and evaluation. J. Hydrol. 505, 47-64. doi:10.1016/j.jhydrol.2013.09.006 McGuire, K. J., & McDonnell, J. J. (2006). A review and evaluation of catchment transit time modeling. Journal of Hydrology, 330(3–4), 543–563. https://doi.org/10.1016/j.jhydrol.2006.04.020 Rinaldo, A., Benettin, P., Harman, C. J., Hrachowitz, M., McGuire, K. J., van der Velde, Y., et al. (2015). Storage selection functions: A coherent framework for quantifying how catchments store and release water and solutes. Water Resources Research, 51, 4840–4847. https://doi.org/10.1002/2015WR017273

---

## Referee Comment (RC3) · Anonymous Referee #3 · 26 Jul 2018

The discussion paper "Real-time observations of stable isotope dynamics during rainfall and throughfall events" presents an suitable approach for continuous observation of stable water isotope composition in precipitation and throughfall at the plot level. The paper is very well structured and written. The presentation of results is in general very clear and straight forward. I think, however, that the discussion could be extended in order to explain the findings or show that with the given data no explanation is possible. The focus of the paper is to test the methodological approach. But as the approach was tested in a natural environment, possible natural influences should be discussed e.g. the possible effect of different initial rainfall $\delta18O$ values for different events and and its possible relation to low isotopic ratios for events with antecedent dry conditions. This could be linked to the level of rainout of the air mass before the events.

[Figure]

The discussion also lacks relating the results to the existing literature. Some examples are given below, but the revision should not be restricted to these examples. Right now, the discussion only provides two citations and one of it is a self-citation. Beside this, the following specific comments should be addressed before the publication of the manuscript in HESS.

Page 2, Line 5: "These effects": The before mentioned effects include "mixing with water from previous events", which does not affect the mean amount of weekly sampled TF compared to Pg. Please improve the wording.

Line7: Not the absolute interception loss is higher for small events, but the relative loss compared to rainfall.

Line 8: Insert "depth" after TF at the end of the line.

Line 13: "isotopic composition" please indicate whether this is related to TF or something else?

Line 16: "94 gauges" These are TF collectors?

Line 18: Change "temporal" to "spatiotemporal".

Page 3, Methods: The bulk sampling method needs to be added to the methods section.

Line 32: Delete "when measured below the canopy" as TF can't be measured elsewhere than below the canopy.

Page 4, Line 7: "periodical back flushing" How often was this done?

Line 17: As it hasn't be said until this point that there was 1 collector for rainfall and one for throughfall the reader can't understand to what "each collector" is referring to. Please clarify.

Line 23-25: Were all these meteorological variables used in the publication? If not than

keep only those that were used.

Page 5, Line 20: "high rainfall intensities" Consider rewording as the reader does not know if "high" is related to this event or to all events sampled.

Line 22-23: Please reword as it is not the samples that are shown but the isotopic ratios of these samples.

Page 6, Line 4: Change "compared" to "correlated to each other".

Line 5: The authors might add that the correlation is significant, but rather moderate.

Line 6: Insert "percentage of" before "interception loss" as the absolut interception loss is probably higher for greater rainfall intensities.

Line 8: There is no positive correlation. The weak correlation is not significant.

Line 13: Please rephrase this sentence. "2-2.5 per mille" were not found "in $\delta$18O values".

Line 17: How was dryness or wetness of the canopy before an event started monitored? I assume that dry and wet is interfered from a certain period of time without rainfall. This should be described in the revised manuscript.

Line 23: If the information about $\Delta\delta$2H is important then I suggest to take it out of the brackets and include it in the sentence with "and" and "respectively". Otherwise it should be omitted.

Page 7, Line 11-12: Is this in line with the literature? E.g. see (Qu et al., 2014). Please extend the discussion.

Line 21-22: Figure 3 only shows one significant correlation. Please correct this sentence.

Line 21: An explanation should be given why there is a negative correlation of rainfall intensity with interception loss. I can imagine that this is in line with what others have

found for bulk samples. If possible respective references should be added.

Line 21-22: Is this in line with results reported in the literature? See e.g. (Dewalle and Swistock, 1994; Kato et al., 2013)

Line 22: I don't understand what the authors mean with "There is no clear pattern for only one of these variables…"

Line 26-27: This needs to be related to what was reported earlier in the literature (e.g. Allen et al., 2013). There is more room for interpretation here. For instance, the difference of $\delta$18O in rainfall and throughfall is presented for different events and discussed, but the level of $\delta$18O of rainfall for different events is not presented. For events for which the degree of rainout from Ocean to inland is low, levels of $\delta$18O of rainfall could be higher. Would it be possible that high initial levels $\delta$18O in rainfall lead to rather smaller increases of $\delta$18O from rainfall to throughfall?

Page 8, Line 7-9: Scatterplots only presented for bulk samples.

Figure 1: For me it is not absolutely clear whether Pg is collected by the same collector for the water that goes through the tipping bucket and that one that is sampled. So is there only one collector for each, Pg and TF, and the water go first through the tipping bucket and then the pump samples the water? Please clarify this in the figure. I think that the triangle within the box illustrates the tipping bucket, right? This must be shown more clearly.

Figure 2: Add "depth" after "Time series of rainfall".

Figure 2: In the text it says that the event duration was only 2 hours, but in the legend a "three-hour bulk sample" is mentioned.

Figure 2: Please provide the date of this event in the legend.

Figure 3: I don't the point in part a) and b) of figure 3. The only additional information is the p-value. But the p-value should be added to the left side figure part for all corre-
lations, e.g. in parentheses below the correlation coefficients. If the reason for a) and b) was to have interception on the x-axis, then it should be plotted as the first variable in the left side figure.

Figure 3: This is not a scatter plot of throughfall samples. Please rephrase the figure caption.

Figure 3: Please indicate in the figure that the intensity is that one of rainfall (and not throughfall).

Figure 3: Please explain in the legend what is shown the lower left und the upper right part of the figure on the left side. Include an explanation, why the size of the correlation coefficients differ. Is that really needed? The small numbers are hard to read.

Figure 3: I suggest adding $\delta$18O of rainfall and throughfall and the length of the antecedent dry period to the scatter plot.

Figure 4: Please shift the right figure to the left side as it is mentioned first in the text.

Figure 5: Which of these events are shown in figures 2 and 6?

Figure 5: There are events for which $\Delta\delta$18O increases with time and others with an opposite trend. Does $\Delta\delta$18O correlate with $\delta$18O of rainfall per event?

Figure 6: Indicate the date of this event.

Figure 6: Delete "amounts," from the first line of the figure caption.

Figure 6: The colours of d_Pg and d_TF are hard to distinguish.

Figure 6: Why did throughfall start before rainfall?

References:

Allen, S.T., Brooks, J.R., Keim, R.F., Bond, B.J., McDonnell, J.J., 2013. The role of pre‐event canopy storage in throughfall and stemflow by using isotopic tracers. Ecohydrology 7, 858–868. https://doi.org/10.1002/eco.1408

Dewalle, D.R., Swistock, B.R., 1994. Differences in oxygen-18 content of throughfall and rainfall in hardwood and coniferous forests. Hydrol. Process. 8, 75–82. https://doi.org/10.1002/hyp.3360080106

Kato, H., Onda, Y., Nanko, K., Gomi, T., Yamanaka, T., Kawaguchi, S., 2013. Effect of canopy interception on spatial variability and isotopic composition of throughfall in Japanese cypress plantations. J. Hydrol. 504, 1–11. https://doi.org/10.1016/j.jhydrol.2013.09.028

Qu, S., Zhou, M., Shi, P., Liu, H., Bao, W., Chen, X., 2014. Differences in oxygen-18 and deuterium content of throughfall and rainfall during different flood events in a small headwater watershed. Isotopes Environ. Health Stud. 50, 52–61. https://doi.org/10.1080/10256016.2014.845565

---

## Author Comment (AC2) · 21 Aug 2018

Response to reviewer comments RC2

We thank you for your thoughtful comments and making us aware of open questions. Please find below a list of specific responses to the individual points.

The manuscript "Real-time observations of stable isotope dynamics during rainfall and throughfall events" by Herbstritt et al. presents and discusses an experimental setup designed to monitor in parallel the stable isotope composition of rainfall (Pg) and throughfall (TF) at high resolution during several summer rainfall events. High resolution stable isotopes in rainfall have already been observed and documented in previous studies. Yet, this study is the first I am aware of that compared the rapid dynamics of

[Figure]

rainfall and throughfall and looked into their mass weighted average difference over several rainfall events. In the abstract and the introduction, the authors summarize the importance, extent, and main reasons of the difference between isotopic composition of throughfall and that of rainfall. The authors justify the need for a comparison of these signatures at a higher resolution than in the past, which reflects their measurement setup. This is described in the method section and well-illustrated in Figure 1. The results show time series of stable isotopes in TF and Pg, differences between them (dynamics and means), and an attempt to find correlations between such differences and meteorological variables. The discussion essentially deals with some of the limitations of the approach. The manuscript is clear and concise, and the figures are of good quality. Yet, no or only minor mechanistic understanding is provided. It will be a good contribution to isotope hydrology, after having addressed several comments and after some questions are clarified. The introduction and the discussion need to state clearer in what way this measurement setup can provide a more accurate estimate of the isotopic recharge in the catchments for typical applications in isotope hydrology. Why is it not enough to just consider the average mass weighted difference between isotopes in TF and Pg? What is a necessary detail of measurement? One figure that would improve in the manuscript in that regard is the relationship between precipitation intensity and Dd for each measured storm. Are isotopic differences larger for higher intensities during a single event? A more detailed description of the applications of tracers in isotope hydrology is also needed in the introduction. The discussion needs to argue for which application this time-varying difference really is important. For instance, are End Member Mixing Analysis (Hooper et al., 1990), isotope hydrograph separation (Klaus & McDonnell, 2013), or travel time modeling (McGuire & McDonnell, 2006; Rinaldo et al., 2015; Hrachowitz et al., 2016) able to incorporate such high-frequency data and distinguish between Pg and TF? Some clarifications of details in the method sections will be necessary (see details below), as some important information was skipped. Furthermore, the English is somewhat bumpy especially in the first half of the introduction, and should be carefully revised before the manuscript is resubmitted. Eventually, one may

consider to change the manuscript into a technical note, since many of the hydrological aspects are discussed rather briefly and the key contribution is the measurement of high-frequency variations in stable isotopes of Pg and TF and their characteristics. No mechanistic understanding is provided by the manuscript. The main conclusion also starts with the technological aspect.

Specific comments: Abstract line 15 The 4 min time-lag information can be confusing with respect to the 2 sec reading interval. Please make it clearer that the time-lag is the transfer time from the collector to the laser in the instrument. How does dispersion/ diffusion potentially influence this?

Response: We inserted "...from rainfall collector to isotope analyser..." after "...four minutes...". Dispersion in the small funnel must be assumed which is why we calculated a moving average. Dispersion and diffusion can be considered neglectable in the tubing given the small diameters.

page 1: line 1ff. Please tighten up and do a better job in pointing out the relevance of isotopes from here until page 2 line 12.

Response: We rephrased the first part of the introduction as suggested.

line 20 I would omit the word "signature", since the stable isotopes are the tracers, while their signatures are measurements.

Response: Changed as suggested.

line 25 Move the citations (Kendall and McDonnell [. . .]) to line 22 after "water cycle" line 25 "There is. . . hydrology" I would omit that sentence which looks somehow too isolated, see previous comment.

Response: Changed as suggested.

line 26 "residence times" is vague. Is it canopy, soil, or catchment residence times? Please be more precise.
Response: We inserted "catchment".

Line28 New sentence after "Allen et al. . .". Delete "Since" Line 28 Citations after "forested" needed.

Response: We rephrased to "Many studies have used the temporal dynamics in the isotopic composition of precipitation for estimating catchment residence times, but especially in forested catchments interception losses and accompanying isotope effects have a significant impact on the input function (Xu et al., 2014; Stockinger et al., 2015; Allen et al., 2017). The importance of understanding rainfall interception processes is presented in a thorough review by Allen et al. (2017)."

page 2: line 1 must be . . ."Allen et al. (2017)."

Response: Changed as suggested.

line 13 delete "Typically"; found for what? Precip? TF? Runoff? References needed and line 15 "spatial variability in general" is to general. Please elaborate

Response: We rephrased l. 13 - 15 to "High spatial intra- and inter-storm variabilities have been found in depth and isotopic composition of TF. A synthesis study analysed the spatial variabilities of TF from 18 selected studies at a global scale. The study showed that the spatial patterns of TF, when related to leaf area index (LAI) as well as to spatial variability in general, were very heterogeneous and ecosystem dependent (Levia, 2011)."

line 17 Ref needed after "diameters". Replace "They" by "The authors" or "Keim et al."

Response: Ref. added, "They" replaced by "The authors"

page 3: line 18 in-situ

Response: Changed as suggested.

line 24,27 SPACE between numeric value and unit needed

Response: Changed as suggested.

page 4: lines 2-4 What is the dead volume inside smaller funnel just before the pump? How is it made sure that all the water exceeding the pump flowrate Qp is spilled into the bulk sample?

Response: after "...was spilled..." in l. 3 we inserted "... and collected via an additional funnel into a sampling bottle. This overflow was volume-weighted, contributing..." In my perception, if Vd is the dead volume of the smaller funnel (let's assume Vd = 3 mL), then assuming complete mixing, the isotope signature effectively recorded is a moving average of the precipitation, with a time window of length Vd/Qp, i.e. about 36 sec. Please elaborate on this!

Response: We see your point and agree. However, time windows shorter than 90 s yielded quite "noisy" data.

lines 14-16 Were these discrete samples analyzed later in the lab?

Response: Yes. We changed the sentence to "...discrete liquid samples were taken every five minutes at the liquid outlet port of the membrane module and anlysed later in the laboratory."

line 19 Is 10 m really sufficient to make sure that there are no effects of the trees at all on the gross precipitation? Did you see an effect of wind direction etc?

Response: Due to the height where the samplers were installed and the size of the tree the distance was considered sufficient (45°, WMO guidelines). We didn't observe meaningful effects of wind direction.

line 20 How many events were recorded in total? It is never mentioned in the text. It also makes it difficult to follow the results. Add also more details about the events in tables.

Response: Number of recorded events is added as suggested. Table will be added.

[Figure]

page 5: lines 2-9 How was calibration applied? Did you apply an individual correction for each rainfall event based on the 3 measured standards? More details are needed here. Also, how did you ensure that there were no memory effects between standards when measuring them consecutively?

Response: Yes, we applied an individual correction for each rainfall event. l. 5: For clarification after "...each rainfall event..." we added "...until a plateau in the isotope readings was reached ($\sim$ 10 minutes)..."

line 6 What is meant by "long term changes in the membranes"? Please elaborate

Response: Small particles could be removed by back flushing, whereas the built-up of biofilms or mechanical changes of the membrane ("membrane fouling") would change its characteristics over time. We inserted "...e.g. built-up of biofilms or mechanical changes (small cracks, fissures) at the membrane..." after "...long term changes"

line 17 It should be mentioned here already why the VPD is calculated.

Response: We inserted "...to indicate potentially high or low evaporation..." after "...is calculated..."

line 20 What date did the event happened? This also needs to appear in the caption of Figure 2. Why not show directly the comparison between isotopes in Pg and TF in Figure 1, as in Figure 6?

Response: We added the date to the figure and the caption as suggested. In Fig. 2 we intended to show the temporal variability in both isotope ratios (d18O and d2H) during one 'long' (2 h) event without bubbles at the contactor, the continuous readings, the noise reduction by the moving average, and to illustrate the stepwise loss of information with discrete liquid samples and moreover with the one event-based bulk sample.

page 6: line 1 It looks like the isotopes in Pg are getting lighter while rainfall intensities are getting lower. Is that not contradictory with the amount effect?

[Figure]

Response: At the end of this specific rainfall event air temperature dropped which could explain this effect. For clarification we added the temperature data to the figure.

lines 3-4 I suppose the interception loss is (Pg-TF)/Pg*100. It should be stated clearly how you calculated it.

Response: This is correct. We added this equation to chapter 2.2 Analyses.

line 13 Were the interception losses and the Dd18O greater with time and plant growth from May to September? A plot with Dd18O in time during the growing season could be useful here. Any data on LAI?

Response: We had no data on LAI. We looked at the effects of the growing season without finding meaningful relationships.

line 15 Is this mean difference flux weighted? I think this is important to emphasize.

Response: Yes, the mean of each event (Pg and TF separately) was flux weighted and the difference was calculated from the flux weighted means. We added this information to the manuscript.

Line 21 "all events", see above, more information needed

Response: We inserted "nine"

line 29: cm3 should be cm2

Response: Changed as suggested

page 7: lines 1-2 Why is the TF signature more damped than the Pg signature?

Response: We assume that this is due to mixing in the canopy. We rephrased this section to make our point clearer.

Lines 2-4 Maybe it is because of the scale, but it does not seem like the VPD is decreasing on figure 6. Please clarify. Also, why not look at the relationship between VPD, Ta, and time-variable Dd for all events? Some meaningful correlation could exist.

Response: We already looked at these relationships without finding meaningful relations.

lines 10- 11 Some statistics about the differences between continuous measurements and the single liquid samples would be nice here to emphasize that point, even though it looks valid just when looking at the figures. For example, what was the average difference between the single liquid samples and the corresponding moving average values for each event? For all events? Does that vary a lot between events?

Response: Will be added as suggested.

lines 17-18 How are "wet", and "dry" canopies defined?

Response: We defined "dry" by inserting "i.e. 6 hours without rainfall" on p. 6 l. 17.

line 25 So, why are the average differences in bulk samples and continuous samples so different? I think this is a crucial part of the manuscript showing why the measurement protocol proposed here is valuable!

Response: We inserted in l. 25 after "(Fig. 4)" "..., which is not surprising. Bulk samples represent a mean isotopic signature, whereas higher resolution measurements capture the extreme values. Furthermore, in the continuously analysed dataset..."

line 27 What process could explain that a wet canopy leads to an even stronger enrichment?

Response: Will be extended in the discussion accordingly

Figure 1 Is the beginning of the event missed because of the stabilization of T? That info would be nice in the figure caption.

Response: We inserted "...starting after temperature at the contactor was stable".

Figure 5 A legend with the date of each event and the corresponding lines would be nice here.

Response: Will be included.

Figure 6 The points for d_Pg and d_TF in the legend are too small and hard to distinguish. The date of the event is missing.

Response: Will be changed accordingly.

Thanks for the interesting contribution to isotope hydrology! :-)

References: Hooper, R.P., Christophersen, N., Peters, N.E., 1990. Modelling streamwater chemistry as a mixture of soilwater end-members – an application to the Panola Mountain catchment, Georgia, USA. J. Hydrol. 116 (1), 321–343.

Hrachowitz, M., Benettin, P., van Breukelen, B. M., Fovet, O., Howden, N. J. K., Ruiz, L., et al. (2016). Transit timesâËŸAËĞT The link between hydrology and water quality at the catchment scale. Wiley Interdisciplinary Reviews: Water, 3(5), 629–657. https://doi.org/10.1002/wat2.1155

Klaus, J., McDonnell, J.J., 2013. Hydrograph separation using stable isotopes: Review and evaluation. J. Hydrol. 505, 47-64. doi:10.1016/j.jhydrol.2013.09.006

McGuire, K. J., & McDonnell, J. J. (2006). A review and evaluation of catchment transit time modeling. Journal of Hydrology, 330(3– 4), 543–563. https://doi.org/10.1016/j.jhydrol.2006.04.020

Rinaldo, A., Benettin, P., Harman, C. J., Hrachowitz, M., McGuire, K. J., van der Velde, Y., et al. (2015). Storage selection functions: A coherent framework for quantifying how catchments store and release water and solutes. Water Resources Research, 51, 4840–4847. https://doi.org/10.1002/2015WR017273

Please also note the supplement to this comment: https://www.hydrol-earth-syst-sci-discuss.net/hess-2018-301/hess-2018-301-AC2-supplement.pdf

---

## Author Comment (AC3) · 21 Aug 2018

Response to reviewer comments RC3

We thank you for your thoughtful comments and making us aware of open questions. Please find below a list of specific responses to the individual points.

The discussion paper "Real-time observations of stable isotope dynamics during rainfall and throughfall events" presents an suitable approach for continuous observation of stable water isotope composition in precipitation and throughfall at the plot level. The paper is very well structured and written. The presentation of results is in general very clear and straight forward. I think, however, that the discussion could be extended in order to explain the findings or show that with the given data no explanation is possible.

[Figure]

The focus of the paper is to test the methodological approach. But as the approach was tested in a natural environment, possible natural influences should be discussed e.g. the possible effect of different initial rainfall d18O values for different events and and its possible relation to low isotopic ratios for events with antecedent dry conditions. This could be linked to the level of rainout of the air mass before the events. The discussion also lacks relating the results to the existing literature. Some examples are given below, but the revision should not be restricted to these examples. Right now, the discussion only provides two citations and one of it is a self-citation. Beside this, the following specific comments should be addressed before the publication of the manuscript in HESS.

Page 2, Line 5: "These effects": The before mentioned effects include "mixing with water from previous events", which does not affect the mean amount of weekly sampled TF compared to Pg. Please improve the wording.

Response: We inserted "...and the difference in isotopic composition..." after "mean amount"

Line7: Not the absolute interception loss is higher for small events, but the relative loss compared to rainfall.

Response: We rephrased the sentence to "...the volume weighted mean of the interception loss..."

Line 8: Insert "depth" after TF at the end of the line.

Response: Changed as suggested.

Line 13: "isotopic composition" please indicate whether this is related to TF or something else?

Response: We inserted "...of TF".

Line 16: "94 gauges" These are TF collectors?

[Figure]

Response: We replaced "gauges" by "TF collectors".

Line 18: Change "temporal" to "spatiotemporal".

Response: Changed as suggested.

Page 3, Methods: The bulk sampling method needs to be added to the methods section.

Response: We rephrased l. 2-4. It now reads: "From there, a stream of water was pumped to the membrane contactor with a peristaltic pump at a constant flowrate of 5 mL/min while at the same time water exceeding this flowrate was spilled and collected via an additional funnel into a sampling bottle. This overflow was volume-weighted, contributing to the event-based bulk sample."

Line 32: Delete "when measured below the canopy" as TF can't be measured elsewhere than below the canopy.

Response: Changed as suggested.

Page 4, Line 7: "periodical back flushing" How often was this done?

Response: We rephrased the sentence to: "... could be facilitated by back flushing with deionized water as needed or periodical rinsing (every 2 to 4 weeks) with weak acids, respectively."

Line 17: As it hasn't be said until this point that there was 1 collector for rainfall and one for throughfall the reader can't understand to what "each collector" is referring to. Please clarify.

Response: Thanks, we replaced "each" by "the".

Line 23-25: Were all these meteorological variables used in the publication? If not than keep only those that were used.

Response: Changed as suggested.

Page 5, Line 20: "high rainfall intensities" Consider rewording as the reader does not know if "high" is related to this event or to all events sampled.

Response: Will be rephrased.

Line 22-23: Please reword as it is not the samples that are shown but the isotopic ratios of these samples.

Response: We inserted ". . .isotope ratios of discrete liquid, . . ." in l. 21 and replaced "sampling" by "analysing" in l. 23.

Page 6, Line 4: Change "compared" to "correlated to each other".

Response: Changed as suggested.

Line 5: The authors might add that the correlation is significant, but rather moderate.

Response: Added as suggested.

Line 6: Insert "percentage of" before "interception loss" as the absolute interception loss is probably higher for greater rainfall intensities.

Response: Changed as suggested.

Line 8: There is no positive correlation. The weak correlation is not significant.

Response: Rephrased as suggested.

Line 13: Please rephrase this sentence. "2-2.5 per mille" were not found "in d18O values".

Response: We rephrased the sentence to "The data of the bulk samples (Fig. 4, right) were grouped into 0.5‰classes. The maximum of 2 - 2.5‰ in Dd18O values was calculated only for two events, while for 23 of the 28 events Dd18O was 1.5‰ or less."

Line 17: How was dryness or wetness of the canopy before an event started monitored? I assume that dry and wet is interfered from a certain period of time without

rainfall. This should be described in the revised manuscript.

Response: We inserted "..., i.e.after at least 6 hours without rainfall..." after "dry canopy"

Line 23: If the information about Dd2H is important then I suggest to take it out of the brackets and include it in the sentence with "and" and "respectively". Otherwise it should be omitted.

Response: Changed as suggested.

Page 7, Line 11-12: Is this in line with the literature? E.g. see (Qu et al., 2014). Please extend the discussion.

Response: Will be extended in the discussion accordingly.

Line 21-22: Figure 3 only shows one significant correlation. Please correct this sentence.

Response: Rephrased as suggested.

Line 21: An explanation should be given why there is a negative correlation of rainfall intensity with interception loss. I can imagine that this is in line with what others have found for bulk samples. If possible respective references should be added.

Response: Will be extended in the discussion accordingly.

Line 21-22: Is this in line with results reported in the literature? See e.g. (Dewalle and Swistock, 1994; Kato et al., 2013)

Response: Will be extended in the discussion accordingly.

Line 22: I don't understand what the authors mean with "There is no clear pattern for only one of these variables. . ."

Response: We rephrased the sentence. It now reads "The explained variance by any of the considered variables alone was generally small, illustrating the complexity of the
processes contributing to interception loss and the transformation of Pg isotope ratios when becoming TF."

Line 26-27: This needs to be related to what was reported earlier in the literature (e.g. Allen et al., 2013).

Response: We rephrased to "This allows for the interpretation that antecedent conditions have a clear impact on isotopic enrichment of TF as also described in previous studies (Keim et al., 2005; Allen et al., 2014; Stockinger et al., 2015; Allen et al., 2017)."

There is more room for interpretation here. For instance, the difference of d18O in rainfall and throughfall is presented for different events and discussed, but the level of d18O of rainfall for different events is not presented.

Response: Added as suggested.

For events for which the degree of rainout from Ocean to inland is low, levels of d18O of rainfall could be higher. Would it be possible that high initial levels d18O in rainfall lead to rather smaller increases of d18O from rainfall to throughfall?

Response: We see your point but currently can't think of a physical reason that would support this hypothesis. We additionally looked at the relationship without finding any significant correlation.

Page 8, Line 7-9: Scatterplots only presented for bulk samples.

Response: We refer to the results shown in Fig. 4 to 6.

Figure 1: For me it is not absolutely clear whether Pg is collected by the same collector for the water that goes through the tipping bucket and that one that is sampled. So is there only one collector for each, Pg and TF, and the water go first through the tipping bucket and then the pump samples the water? Please clarify this in the figure. I think that the triangle within the box illustrates the tipping bucket, right? This must be shown more clearly.

Response: Changed in Figure 1 as suggested.

Figure 2: Add "depth" after "Time series of rainfall".

Response: Changed as suggested.

Figure 2: In the text it says that the event duration was only 2 hours, but in the legend a "three-hour bulk sample" is mentioned.

Response: In order to also collect the water dripping from the canopy, the throughfall sampler was emptied later. We changed the legend to "bulk sample (event)".

Figure 2: Please provide the date of this event in the legend.

Response: Added as suggested.

Figure 3: I don't the point in part a) and b) of figure 3. The only additional information is the p-value. But the p-value should be added to the left side figure part for all correlations, e.g. in parentheses below the correlation coefficients. If the reason for a) and b) was to have interception on the x-axis, then it should be plotted as the first variable in the left side figure.

Response: p-values are added for all correlations as suggested.

Figure 3: This is not a scatter plot of throughfall samples. Please rephrase the figure caption.

Response: Changed as suggested.

Figure 3: Please indicate in the figure that the intensity is that one of rainfall (and not throughfall).

Response: Changed as suggested.

Figure 3: Please explain in the legend what is shown the lower left und the upper right part of the figure on the left side. Include an explanation, why the size of the correlation coefficients differ. Is that really needed? The small numbers are hard to read.

Response: Explanation in the legend is added as suggested. The size of the correlation coefficients is changed (all the same size).

Figure 3: I suggest adding d18O of rainfall and throughfall and the length of the antecedent dry period to the scatter plot.

Response: d18O data are added to the scatter plot as suggested. We didn't show the data about the length of the antecedent dry periods because not all relevant time scales were sufficiently represented in our dataset.

Figure 4: Please shift the right figure to the left side as it is mentioned first in the text.

Response: Changed as suggested.

Figure 5: Which of these events are shown in figures 2 and 6?

Response: We added a legend to figure 5.

Figure 5: There are events for which Dd18O increases with time and others with an opposite trend. Does Dd18O correlate with d18O of rainfall per event?

Response: We looked at this relationship without finding a significant correlation

Figure 6: Indicate the date of this event.

Response: Changed as suggested.

Figure 6: Delete "amounts," from the first line of the figure caption.

Response: Changed as suggested.

Figure 6: The colours of d_Pg and d_TF are hard to distinguish.

Response: Will be changed in the figure.

Figure 6: Why did throughfall start before rainfall?

Response: Rainfall started before 19:27 already, but data is shown from the time when

temperature at the contactor was stable.

References: Allen, S.T., Brooks, J.R., Keim, R.F., Bond, B.J., McDonnell, J.J., 2013. The role of preâËŸARËĞ event canopy storage in throughfall and stemflow by using isotopic tracers. Ecohydrology 7, 858–868. https://doi.org/10.1002/eco.1408

Dewalle, D.R., Swistock, B.R., 1994. Differences in oxygen-18 content of throughfall and rainfall in hardwood and coniferous forests. Hydrol. Process. 8, 75–82. https://doi.org/10.1002/hyp.3360080106

Kato, H., Onda, Y., Nanko, K., Gomi, T., Yamanaka, T., Kawaguchi, S., 2013. Effect of canopy interception on spatial variability and isotopic composition of throughfall in Japanese cypress plantations. J. Hydrol. 504, 1–11. https://doi.org/10.1016/j.jhydrol.2013.09.028

Qu, S., Zhou, M., Shi, P., Liu, H., Bao, W., Chen, X., 2014. Differences in oxygen-18 and deuterium content of throughfall and rainfall during different flood events in a small headwater watershed. Isotopes Environ. Health Stud. 50, 52–61. https://doi.org/10.1080/10256016.2014.845565

Please also note the supplement to this comment:
https://www.hydrol-earth-syst-sci-discuss.net/hess-2018-301/hess-2018-301-AC3-supplement.pdf

---

## Author Response (AR1)

In this manuscript, the authors present results of a novel method to determine stable isotope ratios of H and O (delta2H and delta18O values) in water of incident rainfall and throughfall below a selected individual tree in high temporal resolution making use of the latest developments in infrared laser spectroscopy.

Overall, the conducted research is sound and the manuscript is well structured. However, the language is sloppy and imprecise and needs to be considerably improved.

In the following, I offer a number of line-by-line comments to improve the manuscript before it can be accepted for publication in Hydrology and Earth System Sciences:

Comment: p. 1, l. 1-2: The title is misleading and incomplete. The measurement is not real-time but highly resolved (but with a temporal delay), the considered stable isotope ratios must be specified, because rainfall and throughfall do not only consist of water but include numerous solutes, which also have to a large part several stable isotopes. Therefore, the title should be something like "Temporally highly resolved measurement of stable hydrogen and oxygen isotope ratios in water of rainfall and throughfall with a novel infrared laser-based method"

*Response: We agree that the stable isotopes have to be specified. Therefore we now inserted "water" before "…stable isotope dynamics…". Regarding time delay we agree that strictly speaking the isotope measurements are not real-time. However, when comparing the time delay of our method (few minutes) to the time delay accompanying assays based on discrete liquid samples (days, even weeks depending on the number of samples and laboratory capacities) we considered it justified to call it "real-time". Further, we feel that this should be seen in the context of the research conducted. Ultimately, our setup is intended to be employed in combination with observations of soil or runoff water which will eventually carry the isotopic signature first observed in gross precipitation and canopy throughfall.*

*Compared to the timescale that can be expected for these reactions we argue that "real-time" is not too far-fetched.*

Comment: p. 1, l. 6: Like in the title the considered isotopes and their molecule need to be mentioned.

*Response: We inserted "water" before "…isotopic composition…" in this rather general introductory sentence. Further, we specified the molecule and isotopes in line 10. The sentence now reads "For the quasi real-time observation of the water isotopic composition ($\delta\ ^{18}O$ and $\delta^2H$) of…".*

Comment: p. 1, l. 6: It is unclear what the difference between exchange and mixing is. Add an explanation.

*Response: We refer to diffusive exchange between liquid water and ambient vapour, while mixing refers to conservative mixing of different liquid water reservoirs.*

*We inserted "diffusive" before "…exchange…".*

Comment: p. 1, l. 8 (and throughout the manuscript): It is unclear what you mean by "amount". Do you talk about the rainfall/throughfall rate or volume? Be clear.

*Response: We refer to depth. We found that amount and depth are used synonymously in literature. Nonetheless, we replaced "amount" by "depth" in most places of the manuscript.*

Comment: p. 1, l. 10: What do you mean by "gross precipitation"? The water of the incident rainfall?

*Response: Gross precipitation is the community-used term for open site rainfall.*

Comment: p. 1, l. 16-18: This threeline statement has (almost) no content. Specify, what exactly makes your method to a tool for more insight.

*Response: We rephrased the sentence. It now reads:*

*"The achieved evolution from discrete liquid or event-based bulk samples to continuous measurements allows for direct comparison with common meteorological measurements. This makes our approach a powerful tool towards more insight into the dynamic processes contributing to interception during rainfall events."*

Comments: p. 1, l. 20: Tracers of what? Water sources? Water flow paths? Mixing processes? All of them? But then it would no longer be ideal, because usually a tracer is

expected to be specific for a single source or a single process. & p. 1, l. 21-22: This is again too unspecific. What exactly were water isotopes used for?

*Response p.1 l. 20-22: p. 1, l. 20: This is an introductory sentence. We kept it but rephrased the following sentence (p. 1, l. 21-22). It now reads:*

*"They have proven to be powerful tools for the characterisation of water flow and transport processes with a long record of applications at different spatial and temporal scales and in all parts of the water cycle (Kendall and McDonnell, 1998; Vitvar et al., 2005)."*

Comment: p. 1, l. 24: What is crucial? The knowledge of the isotope ratios? But there was also a catchment hydrology before the isotopes could be measured and there are catchment hydrologists who do not use water isotope ratios.

*Response: We rephrased the sentence to:*

*"The isotopic composition of precipitation ultimately cascades through the entire hydrologic system affecting soil water, groundwater, evapotranspiration, and stream water isotopic signatures. Knowledge about the isotopic composition of precipitation is therefore crucial for isotope studies in catchment hydrology."*

Comment: p. 1, l. 27: "important role" for what? Location, rate and volume of water input into the soil?

*Response: we replaced "…play an important role…" by "…have a significant impact on the input function…"*

Comment: p. 2, l. 1: The parentheses should be around the year only.

*Response: Changed as suggested.*

Comment: p. 2, l. 3: Why should "redistribution in the canopy" have an isotope effect?

*Response: We meant to express that redistribution, i.e. movement of water to or from a specific place e.g. via flow along branches, may have an effect on the isotopic composition observed as it also changes the (unknown) spatial pattern of precipitation water isotopes. We addressed this issue in the discussion where it now reads:*

*"Water redistribution in the canopy, i.e. movement of water to or from a specific place e.g. via flow along branches, may have an effect on the isotopic composition observed as it also changes the unknown spatial pattern of precipitation water isotopes."*

Comment: p. 2, l. 5: The mixing was already mentioned in l. 4.

*Response: The study cited in l. 4 refers to intra-storm mixing, while the study cited in l. 5 refers to mixing with water from previous events.*

Comments: p. 2, l. 6: Perhaps better "deciduous and coniferous" in context with "type", because spruce and beech are plant species. & p. 2, l. 7: To what do the numbers refer? Interception loss? Throughfall in % of the rainfall? Vegetation cover? & p. 2, l. 8: Enrichment of which isotopes in which compound?

*Response p. 2, l. 6-8: We replaced "type" by "species" and rearranged the sentence to:*

*"Depending on the species (spruce and beech) and on the density of the vegetation cover the volume weighted mean of the interception loss was in a range of 12% to 41%. It was typically higher for small events and therefore generally led to enrichment of heavy isotopes in TF (Brodersen et al., 2000)."*

Comment: p. 2, l. 8-10: Is this really done? The following sentence is much more plausible.

*Response: The authors found significant differences between $P_g$ and TF depth in their study and argued not to ignore the differences.*

*We replaced "…is…" by "…would…be…"*

Comment: p. 2, l. 13-15: This sentence is confusing. I do not understand it.

*Response: We rephrased the sentence to:*

*"High spatial intra- and inter-storm variabilities have been found in depth and isotopic composition of TF. A synthesis study analysed the spatial variabilities of TF from 18 selected studies at a global scale. The study showed that the spatial patterns of TF, when related to leaf area index (LAI) as well as to spatial variability in general, were very heterogeneous and ecosystem dependent (Levia, 2011)."*

Comment: p. 2, l. 16-18: The temporal persistence of spatial throughfall patterns clearly depends on the length of the observation period and on the vegetation type. Please specify both. Furthermore, it is unclear what the cited authors studied: throughfall volume, rate or water isotope composition?

*Response: Regarding the vegetation type we added "(coniferous and deciduous)"*

*The cited authors studied throughfall amounts (l. 17), we added "…(storm-total) for three to seven storms in a six months period..." before "…with a geostatistical approach…"*

Comments: p. 2, l. 20: Are you talking about the same study that you cited just before? Who "hypothesized"? You? & p. 2, l. 21: The collection of representative throughfall volume/rate

data is a classic in ecosystem sciences and it is rather well known how it can be reached. See e.g., Kimmins, J.P., 1973. Some statistical aspects of sampling throughfall precipitation in nutrient cycling studies in British Columbian coastal forests. Ecology 54, 1008–1019. Do you refer to representative stable water isotope ratios?

*Response p. 2, l. 20-22: We moved the citation from l. 19 to l.22*

*The authors referred to stable isotope ratios, therefore we added "… for isotope studies…" before "…is still missing…"*

Comment: p. 3, l. 12-14: However, if you want to investigate the mean water isotope ratios of throughfall, you also need a high spatial resolution of your samples to collect a representative throughfall sample as stated earlier in your introduction. Just one pair of samplers above and below the canopy is not sufficient for this purpose.

*Response: We are aware of this issue and fully agree. However, we argue that before a setup for TF sampling can be employed multiple times in the field in order to cover the spatial variability of TF isotope patterns the technical challenges need to be solved first. The present paper is meant to aim at this goal.*

Comment: p. 3, l. 24-25: Perhaps, the explanation can be a little bit expanded to avoid that the reader has to consult the cited paper.

*Response: The method of the cited paper is summarized in l. 19-29. For clarification we modified the sentence in l. 24-25 to*

*At the membrane's surface, dry carrier gas (e.g. $N_2$) mixes with vapour diffusing through the pores across the membrane from the liquid phase. Moist air then leaves the contactor…*

Comment: p.3 l. 30: The spatial variability of throughfall cannot be appropriately measured with a single collector of limited size. Usually, a large number of collectors is needed to collect a representative sample. I suggest that you make clear that your results refer to a point measurement, which is very likely not representative for the throughfall at larger scale.

*Response:*

*In this study we did not intend to investigate the spatial variability. Therefore, we modified the last paragraph of the introduction to:*

*"Therefore, the aim of this study is to develop an approach for the analysis of $P_g$ and TF depth and isotopic composition at the point level at high temporal resolution based on the membrane contactor method, to compare and validate the continuous isotope measurements with discrete liquid samples as well as with event-based bulk samples. With this approach the dynamics in amount and isotopic composition of $P_g$ and TF and hence, interception processes influencing amount and isotopic composition of TF can be investigated in unprecedented high temporal resolution."*

Comment: p. 4, l. 19: Where was incident rainfall measured? Above the canopy or in an adjacent forest clearing?

*Response: We replaced "...10 m apart..." by "...with 10 m horizontal distance from each other..."*

Comment: p. 4, l. 20: Acer campestre (in italics) L. - i.e. spell genus name with upper scale A and include author Abbreviation.

*Response: Changed as suggested.*

Comment: p. 5, l. 10: What does this deviation from the meteoric line tell us? What is the reason for nonequilibrium fractionation? The d value only appears in Fig. 6 but is not addressed in the discussion.

*Response: The d values are now considered in the discussion. The respective paragraph reads:*

*"The lower intensity and total depth of TF as compared to $P_g$ is indicative for evaporation from the canopy. However, this is not reflected in continuous $d_{TF}$ values (Fig. 6) which were expected to follow the trend of $d_{Pg}$ values while being lower after evaporative non-equilibrium fractionation."*

Comment: p. 5, l. 14 (and troughout the manuscript): Capdelta values are given without the lower case delta (i.e. DELTA18O, not DELTAdelta18O).

*Response: Which guideline do you refer to? DELTAdelta18O is mathematically correct and it is common in the community. Therefore we defined it like this in Eq. (2) and (3).*

Comment: p. 5, l. 20: Why did you chose exactly this event? Add properties of the event (date, total volume, intensity).

*Response: We added the date to the figure and the caption as suggested. In Fig. 2 we intended to show the temporal variability in both isotope ratios ($\delta^{18}O$ and $\delta^{2}H$) during one 'long' (2 h) event without bubbles at the contactor, the continuous readings, the noise reduction by the moving average, and to illustrate the stepwise loss of information with discrete liquid samples and moreover with the one event-based bulk sample.*

Comment: p. 5, l. 21: "grab" samples, anyway being sloppy jargon, sounds strange in the context of water. I at least cannot grab water...

*Response: We replaced "… liquid grab…" by "… isotope ratios of discrete…". Throughout the manuscript we replaced "liquid grab samples" by "discrete liquid samples".*

Comment: p. 6, l. 13: Usually, one starts with the left figure and then goes to the right one.

*Response: Changed as suggested.*

Comment: p. 6, l. 29: "cm3" is not the unit of an area.

*Response: Changed to cm².*

Comment: p. 6, l. 33: "without bubbles at any contactor"

*Response: Changed as suggested.*

Comment: p. 7, l. 11-13: This statement seems almost trivial. What do we really gain by this higher temporal resolution? How do you interpret Fig. 2? I am a bit disappointed of the depth of the discussion.

*Response: We think that the value of our study is, that the high temporal resolution achieved now matches the resolution of routine meteorological observations and thus allows for comparison with these. Consequently, as explained in the revised discussion, continuous $P_g$ and TF isotope data enable new mechanistic modeling approaches aiming at a more realistic simulation of physical processes being effective when $P_g$ becomes TF.*

*Figure 2 is now addressed several times in the revised discussion:*

*"The modified setup of the method developed by Herbstritt et al. (2012) adapted to continuous rainfall and throughfall isotope measurement worked quite well in terms of providing continuous, thermo-regulated flows of water to the membrane modules and delivering reliable liquid water stable isotope data. The latter became evident by the good agreement of continuous measurements and single liquid samples (**Fig. 2** and also Fig. 6) which was in the order of the measurement uncertainty for both isotope ratios under investigation. Large intra-storm variabilities exist in the isotopic signature of $P_g$ and TF, which would have been impossible to detect when solely relying on commonly taken event-based bulk samples or even on data representing a higher sampling interval of typically 5 mm precipitation depth. We found that the variability of the continuous $P_g$ isotope data is higher than the variability of the continuous TF data (Fig. 6). For the latter, the signal is more dampened probably due to mixing processes, which was also found in other studies (Qu et al., 2013). The lower intensity and total depth of TF as compared to $P_g$ is indicative for evaporation from the canopy. However, this is not reflected in continuous $d_{TF}$ values (Fig. 6) which were expected to follow the trend of $d_{Pg}$ values while being lower after evaporative non-equilibrium fractionation."*

*…*

*"Depletion in heavy isotopes in open site rainfall was observed in the last ~30 minutes of the event presented in **Figure 2**. Several effects could cause such a pattern. The amount effect, reported for lower latitudes (Moore et al., 2014), can be ruled out due to the fact that at the same time rainfall intensity was quite low compared to other periods of this particular rainfall event. Also a rainout effect cannot be attributed to our data as it is only detectible on a spatial scale considering the movement of air masses and rain clouds. Rather, the simultaneous decrease of air temperature indicates the passing of a weather front which appears to be the relevant explanation for the observed changes in precipitation isotope values."*

*…*

*"Highest $\Delta\delta^{18}O$ values were found in the cases of the continuous observations. One reason could be that the continuously analysed events are shorter and therefore potentially capturing extreme values while bulk samples represent flux-weighted mean isotopic signatures of the entire periods of rainfall and throughfall. On the other hand, we are aware that the calculation and interpretation of synchronous $\Delta\delta^{18}O$ and $\Delta\delta^{2}H$ data are disputable given assumable, yet unknown, time lags between $P_g$ and TF. Within each rainfall event, also past trends and variabilities of the $P_g$ isotopic compositions must be assumed to be reflected in instantaneous TF isotopic compositions, but are not considered with the proposed approach. However, a quite common intra-event $P_g$ isotopic depletion trend (**Fig. 2** and Fig. 6) combined with any positive time lag between $P_g$ and TF would result in higher synchronous $\Delta\delta^{18}O$ values compared to estimates derived from bulk sample data."*

Comment: p. 7, l. 21-22 and Fig. 3: Why do you highlight a non-significant correlation between interception loss and D18O?

*Response: The correlation between interception loss and intensity as well as the correlation between interception loss and $\Delta\delta^{18}O$ were selected due to their relatively high Pearson correlation coefficients and their relatively low p-values. Furthermore we expected to see a significant correlation as it has been shown in the cited literature.*

Furthermore, you should not show a regression line for non-significant correlations.

*Response: We added data to the scatterplot matrix in Fig. 3. Detailed figures with regression lines have been removed.*

Comment: p. 7, l. 22: It is unclear what you mean by "There is no clear pattern…" Perhaps: "The explained variance by any of the considered variables alone was generally small, illustrating the complexity of the processes…"

*Response: Thank you for the suggestion which was followed.*

Comment: p. 7, l. 24-25: This sentence is confusing. It is clear that a bulk sample represents a mean isotopic signature, while at higher resolution the extreme values can be seen, which is trivial.

*Response: We agree that this may appear trivial. The point we are trying to emphasize is that depending on the scale of an isotope study high-resolution data is crucial but was not accessible so far with more traditional approaches. Furthermore this sentence was meant to be rather introductory.*

Comment: p. 8, l. 1-2: Furthermore, there is a pronounced spatial variation of throughfall quantity and quality, which cannot be captured with a single collector (see above).

*Response: We did not intend to cover the spatial variation. We tried to make this point clearer when stating the aims of our study as well as in the revised conclusions. The respective segments now read:*

*"Therefore, the aim of this study is to develop an approach for the analysis of $P_g$ and TF depth and isotopic composition at the point level at high temporal resolution based on the membrane contactor method, to compare and validate the continuous isotope measurements with discrete liquid samples as well as with event-based bulk samples. With this approach the dynamics in amount and isotopic composition of Pg and TF and hence, interception processes influencing amount and isotopic composition of TF can be investigated in unprecedented high temporal resolution."*

*…*

*"We are therefore confident that our setup, especially when employed across larger spatial scales, will contribute to the aim of thorough isotopic sampling of TF, which is crucial in hydrological studies in particular for forested sites but also for other vegetated areas."*

Comment: p. 8, l. 4: This repeats l. 22 (as the whole Fig. 6 is repetitive of Fig. 2 – albeit for another arbitrarily selected rainfall event).

*Response: We added the date to the figure and the caption as suggested. In Fig. 2 we intended to show the temporal variability in both isotope ratios ($\delta^{18}O$ and $\delta^{2}H$) during one 'long' (2 h) event without bubbles at the contactor, the continuous readings, the noise reduction by the moving average, and to illustrate the stepwise loss of information with discrete liquid samples and moreover with the one event-based bulk sample.*

*In Fig. 6 only the moving average is shown, no continuous readings and no bulk sample data. $P_g$- and TF-data of the parallel measurements are shown, including potential artefacts due to bubbles in the contactor when intensities get below a certain threshold. Additionally, meteorological variables are shown that may influence the isotopic signature.*

Comment: p. 8, l. 7: extent

*Response: Changed as suggested.*

Comment: p. 8, l. 11: The ultimate objective would be the prediction of the mean input signal of throughfall into soil for a whole forested catchment or even the spatial distribution of this input signal, which would require much more extensive instrumentation.

*Response: We fully agree, but we did not intend to cover the spatial variation as pointed out above.*

Comment: p. 8, l. 14: I think that "point" level is more appropriate, because you cannot extrapolate your measurement with a single collector to a larger area. Plots I think of are at least 10 x 10 m large and on such plots you might easily need > 10 collectors to measure the mean throughfall properties with an acceptable uncertainty.

*Response: We deleted "…at the plot level…" and instead addressed this issue a few lines later:*

*"We are therefore confident that our setup, especially when employed across larger spatial scales, will contribute to the aim of thorough isotopic sampling of TF, which is crucial in hydrological studies in particular for forested sites but also for other vegetated areas."*

Comment: p. 8, l. 18: I would really be keen to learn about what we can gain by measuring the stable isotope ratios of water at this high resolution. Can we distinguish different processes or even quantify the contributions of these processes to the total throughfall?

*Response: We agree that effort should be made to identify the different processes contributing to total throughfall. We are confident that the method described here has the potential to contribute to this goal. Further, in the revised manuscript we sketched a modeling approach aiming at this goal.*

Comment: p. 8, l. 22: I agree that this is crucial, but again point at the problem of spatial representativity of the measurements which cannot be reached with the approach used in this paper.

*Response: We rephrased the respective sentence which now reads:*

*"We are therefore confident that our setup, especially when employed across larger spatial scales, will contribute to the aim of thorough isotopic sampling of TF, which is crucial in hydrological studies in particular for forested sites but also for other vegetated areas."*

Comment: p. 8, l. 28: I wonder whether the authors are aware of the modelling efforts of Rutter et al. (1971), Agric. Meteorol., Gash and Morton (1978), J. Hydrol. and Gash (1979), Q. J. R. Meteorol. Soc. I think that it could be a way forward to add an isotope module to these models.

*Response: We are aware of these publications and agree that this could be a way forward. Our perception is that the issues of intra-canopy mixing and the time lag between $P_g$ and TF need to be addressed before further modelling efforts are feasible.*

Comment: Fig. 3: Add how many events were sampled to the figure legend. Furthermore, part of the lettering is too small.

*Response: Added and changed as suggested.*

Comment: Fig. 4: Number subfigures.

*Response: Changed as suggested.*

Comment: Fig. 6: I suggest to combine this figure with Fig. 2. Both figures show stable isotope results for arbitrarily chosen individual events but only Fig. 6 is accompanied by the necessary information about the (micro-)meteorologic conditions.

*Response: Both figures are now addressed to a greater extent in the revised discussion. We therefore considered it justified to present them both individually.*

Furthermore, the d values are only shown but not interpreted. Either you add an interpretation of these results or remove the d values entirely.

*Response: The d values are now considered in the revised discussion. The respective paragraph reads:*

*"The lower intensity and total depth of TF as compared to $P_g$ is indicative for evaporation from the canopy. However, this is not reflected in continuous $d_{TF}$ values (Fig. 6) which were expected to follow the trend of $d_{Pg}$ values while being lower after evaporative non-equilibrium fractionation."*

Furthermore, I am confused by the legend stating "in vapour". I understood that you indeed measured isotope ratios in vapour produced from a liquid sample in your contactor but you referred these values back to the liquid sample via a temperature-dependent calibration function. Do you indeed want to show the isotope ratios in vapour (not referred back to the liquid sample)? Why?

*Response: We regret the confusion. You understood right that all vapour data is back calculated to the liquid phase. By referring to 'vapour' in figure caption and legend we meant to indicate where our continuous data were derived from.*

*We added "…derived from…" in the figure caption to avoid this confusion.*

**Response to reviewer comments RC2**

*We thank you for your thoughtful comments and making us aware of open questions.*

*Please find below a list of specific responses to the individual points.*

***Responses to comments are shown in blue.***

Hydrol. Earth Syst. Sci. Discuss.,

https://doi.org/10.5194/hess-2018-301-RC2, 2018

The manuscript "Real-time observations of stable isotope dynamics during rainfall and throughfall events" by Herbstritt et al. presents and discusses an experimental setup designed to monitor in parallel the stable isotope composition of rainfall (Pg) and throughfall (TF) at high resolution during several summer rainfall events. High resolution stable isotopes in rainfall have already been observed and documented in previous studies. Yet, this study is the first I am aware of that compared the rapid dynamics of rainfall and throughfall and looked into their mass weighted average difference over several rainfall events. In the abstract and the introduction, the authors summarize the importance, extent, and main reasons of the difference between isotopic composition of throughfall and that of rainfall. The authors justify the need for a comparison of these signatures at a higher resolution than in the past, which reflects their measurement setup. This is described in the method section and well-illustrated in Figure 1. The results show time series of stable isotopes in TF and Pg, differences between them (dynamics and means), and an attempt to find correlations between such differences and meteorological variables. The discussion essentially deals with some of the limitations of the approach.

The manuscript is clear and concise, and the figures are of good quality.

Yet, no or only minor mechanistic understanding is provided. It will be a good contribution to isotope hydrology, after having addressed several comments and after some questions are clarified.

The **introduction** and the **discussion** need to state clearer in what way this measurement setup can provide a more accurate estimate of the isotopic recharge in the catchments for typical applications in isotope hydrology.

Why is it not enough to just consider the average mass weighted difference between isotopes in TF and Pg?

What is a necessary detail of measurement?

One figure that would improve in the manuscript in that regard is the relationship between precipitation intensity and $\Delta\delta$ for each measured storm. Are isotopic differences larger for higher intensities during a single event?

A more detailed description of the applications of tracers in isotope hydrology is also needed in the introduction. The discussion needs to argue for which application this time-varying difference really is important. For instance, are End Member Mixing Analysis (Hooper et al., 1990), isotope hydrograph separation (Klaus & McDonnell, 2013), or travel time modeling (McGuire & McDonnell, 2006; Rinaldo et al., 2015; Hrachowitz et al., 2016) able to incorporate such high-frequency data and distinguish between Pg and TF?

Some clarifications of details in the method sections will be necessary (see details below), as some important information was skipped.

Furthermore, the English is somewhat bumpy especially in the first half of the introduction, and should be carefully revised before the manuscript is resubmitted.

Eventually, one may consider to change the manuscript into a technical note, since many of the hydrological aspects are discussed rather briefly and the key contribution is the measurement of high-frequency variations in stable isotopes of Pg and TF and their characteristics. No mechanistic understanding is provided by the manuscript. The main conclusion also starts with the technological aspect.

**Specific comments**:

**Abstract**

line 15 The 4 min time-lag information can be confusing with respect to the 2 sec reading interval. Please make it clearer that the time-lag is the transfer time from the collector to the laser in the instrument. How does dispersion/ diffusion potentially influence this?

*Response: We inserted "…from rainfall collector to isotope analyser…" after "…four minutes…". Dispersion in the small funnel must be assumed which is why we calculated a*

*moving average. Dispersion and diffusion can be considered neglectable in the tubing given the small diameters.*

**page 1:** line 1ff. Please tighten up and do a better job in pointing out the relevance of isotopes from here until page 2 line 12.

*Response: We rephrased major parts of the introduction, pointing out the importance of isotope studies for the determination of water flow and transport processes.*

line 20 I would omit the word "signature", since the stable isotopes are the tracers, while their signatures are measurements.

*Response: Changed to "ratios".*

line 25 Move the citations (Kendall and McDonnell [. . .]) to line 22 after "water cycle" line 25 "There is. . . hydrology" I would omit that sentence which looks somehow too isolated, see previous comment.

*Response: Changed as suggested.*

line 26 "residence times" is vague. Is it canopy, soil, or catchment residence times? Please be more precise.

*Response: We inserted "catchment".*

Line28 New sentence after "Allen et al. . .". Delete "Since" Line 28 Citations after "forested" needed.

*Response: We rephrased to*

*"Many studies have used the temporal dynamics in the isotopic composition of precipitation for estimating catchment residence times, but especially in forested catchments, when meteorological and isotopic reference stations are in the open, interception losses and accompanying isotope effects have a significant impact on the input function (Xu et al., 2014;*

*Stockinger et al., 2015; Allen et al., 2017). The importance of understanding rainfall interception processes is presented in a thorough review by Allen et al. (2017)."*

**page 2**: line 1 must be . . ."Allen et al. (2017)."

*Response: Changed as suggested.*

line 13 delete "Typically"; found for what? Precip? TF? Runoff? References needed

and

line 15 "spatial variability in general" is to general. Please elaborate

*Response: We rephrased the respective segment to:*

*"High spatial intra- and inter-storm variabilities have been found in depth and isotopic composition of TF. A synthesis study analysed the spatial variabilities of TF from 18 selected studies at a global scale. The study showed that the spatial patterns of TF, when related to leaf area index (LAI) as well as to spatial variability in general, were very heterogeneous and ecosystem dependent (Levia et al., 2011)."*

line 17 Ref needed after "diameters". Replace "They" by "The authors" or "Keim et al."

*Response: A reference was added. We replaced "They" by "The authors".*

**page 3:** line 18 in-situ

*Response: Changed as suggested.*

line 24,27 SPACE between numeric value and unit needed

*Response: Changed as suggested.*

**page 4**: lines 2-4 What is the dead volume inside smaller funnel just before the pump? How is it made sure that all the water exceeding the pump flowrate Qp is spilled into the bulk sample?

*Response: after "...was spilled…" we inserted*

*"… and collected via an additional funnel into a sampling bottle. This overflow was volume-weighted, contributing…"*

In my perception, if Vd is the dead volume of the smaller funnel (let's assume Vd = 3 mL), then assuming complete mixing, the isotope signature effectively recorded is a moving average of the precipitation, with a time window of length Vd/Qp, i.e. about 36 sec. Please elaborate on this!

*Response: We see your point and agree. However, time windows shorter than 90 s yielded quite "noisy" data.*

lines 14-16 Were these discrete samples analyzed later in the lab?

*Response: Yes, these samples were analyzed later in the lab. For clarification we changed the sentence to*

*"…discrete liquid samples were taken every five minutes at the liquid outlet port of the membrane module and analysed later in the laboratory."*

line 19 Is 10 m really sufficient to make sure that there are no effects of the trees at all on the gross precipitation? Did you see an effect of wind direction etc?

*Response: Due to the height where the samplers were installed and the size of the tree the distance was considered sufficient (45° vertical clearance, WMO guidelines). We did not observe meaningful effects of wind direction.*

line 20 How many events were recorded in total? It is never mentioned in the text. It also makes it difficult to follow the results. Add also more details about the events in tables.

*Response: The number of recorded events is added as suggested:*

*"In total 28 bulk samples and nine continuously analysed events were obtained in August-September 2015 and throughout the vegetation period (May-September) of 2016."*

*A table has been added listing the characteristics of the continuously analysed events.*

**page 5:** lines 2-9 How was calibration applied? Did you apply an individual correction for each rainfall event based on the 3 measured standards? More details are needed here. Also, how did you ensure that there were no memory effects between standards when measuring them consecutively?

*Response: Yes, we applied an individual calibration for each rainfall event. For clarification we added "…until a plateau in the isotope readings was reached (~ 10 minutes)…" after "…each rainfall event…".*

line 6 What is meant by "long term changes in the membranes"? Please elaborate

*Response: Small particles could be removed by back flushing, whereas the built-up of biofilms or mechanical changes of the membrane ("membrane fouling") would change its characteristics over time. We inserted "…e.g. built-up of biofilms or mechanical changes (small cracks, fissures) at the membrane…" after "…long term changes…".*

line 17 It should be mentioned here already why the VPD is calculated.

*Response: We inserted "…to indicate potentially high or low evaporation…" after "…is calculated…".*

line 20 What date did the event happened? This also needs to appear in the caption of Figure 2. Why not show directly the comparison between isotopes in Pg and TF in Figure 1, as in Figure 6?

*Response: We added the date to the figure and the caption as suggested. In Fig. 2 we intended to show the temporal variability in both isotope ratios ($\delta^{18}O$ and $\delta^{2}H$) during one 'long' (2 h) event without bubbles at the contactor, the continuous readings, the noise reduction by the moving average, and to illustrate the stepwise loss of information with discrete liquid samples and moreover with the one event-based bulk sample.*

**page 6:** line 1 It looks like the isotopes in Pg are getting lighter while rainfall intensities are getting lower. Is that not contradictory with the amount effect?

*Response: At the end of this specific rainfall event air temperature dropped, which could explain this effect. For clarification we added the temperature data to the figure. Further, we added the following paragraph to the revised discussion:*

*"Depletion in heavy isotopes in open site rainfall was observed in the last ~30 minutes of the event presented in Figure 2. Several effects could cause such a pattern. The amount effect, reported for lower latitudes (Moore et al., 2014), can be ruled out due to the fact that at the same time rainfall intensity was quite low compared to other periods of this particular rainfall event. Also a rainout effect cannot be attributed to our data as it is only detectible on a spatial scale considering the movement of air masses and rain clouds. Rather, the simultaneous decrease of air temperature indicates the passing of a weather front which appears to be the relevant explanation for the observed changes in precipitation isotope values."*

lines 3-4 I suppose the interception loss is (Pg-TF)/Pg*100. It should be stated clearly how you calculated it.

*Response: This is correct. We added this equation to the method section.*

line 13 Were the interception losses and the $\Delta\delta^{18}O$ greater with time and plant growth from May to September? A plot with $\Delta\delta^{18}O$ in time during the growing season could be useful here. Any data on LAI?

*Response: We looked at potential effects of the vegetation period without finding meaningful relationships. We added the following sentence to the manuscript:*

*"In any case, the measurements were carried out during the period when the leafs had reached their full size, in order to minimize the influence of the growing season."*

*Unfortunately, we did not have any LAI data.*

line 15 Is this mean difference flux weighted? I think this is important to emphasize.

*Response: Yes, the mean of each event ($P_g$ and TF separately) was flux weighted and the difference was calculated from the flux weighted means. We added this information to the manuscript.*

Line 21 "all events", see above, more information needed

*Response: We replaced "…all events…" by "…nine continuously observed events…".*

line 29: cm3 should be cm2

*Response: Changed as suggested.*

**page 7**: lines 1-2 Why is the TF signature more damped than the Pg signature?

*Response: We assume that this is due to mixing in the canopy. This issue is now addressed in the revised discussion:*

*"We found that the variability of the continuous $P_g$ isotope data is higher than the variability of the continuous TF data (Fig. 6). For the latter, the signal is more dampened probably due to mixing processes, which was also found in other studies (Qu et al., 2013)."*

Lines 2-4 Maybe it is because of the scale, but it does not seem like the VPD is decreasing on figure 6. Please clarify. Also, why not look at the relationship between VPD, Ta, and time-variable $\varDelta\delta$ for all events? Some meaningful correlation could exist.

*Response: We looked at these relationships without finding meaningful relations.*

lines 10- 11 Some statistics about the differences between continuous measurements and the single liquid samples would be nice here to emphasize that point, even though it looks valid just when looking at the figures. For example, what was the average difference between the single liquid samples and the corresponding moving average values for each event? For all events? Does that vary a lot between events?

*Response: This information can be found in the results section of the revised manuscript:*

*"Mean absolute deviation between the moving average of the continuously analysed vapour data and the discrete liquid samples was 0.11‰ and 1.35‰ with standard deviations of 0.096‰ and 0.81‰ for $\delta^{18}O$ and $\delta^2H$, respectively."*

*We also extended the discussion by inserting:*

*"…which was in the order of the measurement uncertainty for both isotope ratios under investigation."*

lines 17-18 How are "wet", and "dry" canopies defined?

*Response: We defined "dry" by adding "…i.e. 6 hours without rainfall,…" to the results section.*

line 25 So, why are the average differences in bulk samples and continuous samples so different? I think this is a crucial part of the manuscript showing why the measurement protocol proposed here is valuable!

*Response: We discussed this issue now in greater detail and also provided a possible explanation for the observed discrepancies:*

*"Comparing $\Delta\delta^{18}O$ data during continuously measured events with those derived from event-based bulk samples, the differences are larger during the shorter continuously measured periods (Fig. 4). It should be noted that in this case the definition of a rainfall event is not consistent. Bulk samples cover the entire time period of rainfall regardless of intensity at the point of observation and mere existence of rainfall at the complementary observation point ($P_g$ vs. TF). In contrast, continuous events are defined by sufficient simultaneous rainfall intensity at both points of observation due to our setups' properties. Therefore, natural rainfall events can only partially be captured by continuous synchronous observations. At the same time bulk samples represent the flux-weighted mean of all conditions constituting the respective event."*

*…*

*"Highest $\Delta\delta^{18}O$ values were found in the cases of the continuous observations. One reason could be that the continuously analysed events are shorter and therefore potentially capturing extreme values while bulk samples represent flux-weighted mean isotopic signatures of the entire periods of rainfall and throughfall. On the other hand, we are aware that the calculation and interpretation of synchronous $\Delta\delta^{18}O$ and $\Delta\delta^{2}H$ data are disputable given assumable, yet unknown, time lags between $P_g$ and TF. Within each rainfall event, also past trends and variabilities of the $P_g$ isotopic compositions must be assumed to be reflected in instantaneous TF isotopic compositions, but are not considered with the proposed*

*approach. However, a quite common intra-event $P_g$ isotopic depletion trend (Fig. 2 and Fig. 6) combined with any positive time lag between $P_g$ and TF would result in higher synchronous $\Delta\delta^{18}O$ values compared to estimates derived from bulk sample data."*

line 27 What process could explain that a wet canopy leads to an even stronger enrichment?

*Response: We added the following sentence to the discussion:*

*"The fact that wet canopies lead to an even stronger enrichment in heavy isotopes can probably be attributed to partly evaporated and therefore isotopically enriched pre-event water that mixes with new rainfall water."*

**Figure 1** Is the beginning of the event missed because of the stabilization of T? That info would be nice in the figure caption.

*Response: Yes, the beginning of the event was missed. Therefore, we inserted*

*"…starting after temperature at the contactor was stable".*

**Figure 5** A legend with the date of each event and the corresponding lines would be nice here.

*Response: We added a legend with IDs to every line in Fig. 5. Further, we added a table listing the date and further characteristics of each event.*

**Figure 6** The points for d_Pg and d_TF in the legend are too small and hard to distinguish. The date of the event is missing.

*Response: We changed the color to make the symbols more distinguishable. Further, the date of the event was added to the figure caption.*

Thanks for the interesting contribution to isotope hydrology!

[Figure]
 ☺

The discussion paper "Real-time observations of stable isotope dynamics during rainfall and throughfall events" presents an suitable approach for continuous observation of stable water isotope composition in precipitation and throughfall at the plot level. The paper is very well structured and written. The presentation of results is in general very clear and straight forward. I think, however, that the discussion could be extended in order to explain the findings or show that with the given data no explanation is possible.

The focus of the paper is to test the methodological approach. But as the approach was tested in a natural environment, possible natural influences should be discussed e.g. the possible effect of different initial rainfall d18O values for different events and and its possible relation to low isotopic ratios for events with antecedent dry conditions.

This could be linked to the level of rainout of the air mass before the events.

The discussion also lacks relating the results to the existing literature. Some examples are given below, but the revision should not be restricted to these examples. Right now, the discussion only provides two citations and one of it is a self-citation. Beside this, the following specific comments should be addressed before the publication of the manuscript in HESS.

**Page 2**, Line 5: "These effects": The before mentioned effects include "mixing with water from previous events", which does not affect the mean amount of weekly sampled TF compared to Pg. Please improve the wording.

*Response: Thanks. You are absolutely correct. We accidently mixed up two literature sources. We replaced "…weekly…" by "…event-based…" to be in concordance with Allen et al. (2014).*

Line7: Not the absolute interception loss is higher for small events, but the relative loss compared to rainfall.

*Response: We rephrased the sentence to "…the volume weighted mean of the relative interception loss…"*

Line 8: Insert "depth" after TF at the end of the line.

*Response: Changed as suggested.*

Line 13: "isotopic composition" please indicate whether this is related to TF or something else?

*Response: We inserted "…of TF".*

Line 16: "94 gauges" These are TF collectors?

*Response: We replaced "gauges" by "TF collectors".*

Line 18: Change "temporal" to "spatiotemporal".

*Response: Changed as suggested.*

**Page 3, Methods**: The bulk sampling method needs to be added to the methods section.

*Response: We rephrased the respective segment. It now reads:*
*"From there, a stream of water was pumped to the membrane contactor with a peristaltic pump at a constant flowrate of 5 mL/min while at the same time water exceeding this flowrate was spilled and collected via an additional funnel into a sampling bottle. This overflow was volume-weighted, contributing to the event-based bulk sample."*

Line 32: Delete "when measured below the canopy" as TF can't be measured elsewhere than below the canopy.

*Response: Changed as suggested.*

**Page 4**, Line 7: "periodical back flushing" How often was this done?

*Response: We rephrased the sentence to:*
*"… could be facilitated by back flushing with deionized water as needed or periodical rinsing (every 2 to 4 weeks) with weak acids, respectively."*

Line 17: As it hasn't be said until this point that there was 1 collector for rainfall and one for throughfall the reader can't understand to what "each collector" is referring to. Please clarify.

*Response: Thanks, we replaced "each" by "the".*

Line 23-25: Were all these meteorological variables used in the publication? If not than keep only those that were used.

*Response: Changed as suggested.*

**Page 5**, Line 20: "high rainfall intensities" Consider rewording as the reader does not know if "high" is related to this event or to all events sampled.

*Response: "(Fig. 2)" was added for clarification.*

Line 22-23: Please reword as it is not the samples that are shown but the isotopic ratios of these samples.

*Response: We inserted "…isotope ratios of discrete liquid, …" and replaced "sampling" by "analysing".*

**Page 6**, Line 4: Change "compared" to "correlated to each other".

*Response: Changed as suggested.*

Line 5: The authors might add that the correlation is significant, but rather moderate.

*Response: Added as suggested.*

Line 6: Insert "percentage of" before "interception loss" as the absolute interception loss is probably higher for greater rainfall intensities.

*Response: Changed as suggested.*

Line 8: There is no positive correlation. The weak correlation is not significant.

*Response: Rephrased as suggested.*

Line 13: Please rephrase this sentence. "2-2.5 per mille" were not found "in d18O values".

*Response: We rephrased the sentence to*
*"The data of the bulk samples (Fig. 4, right) were grouped into 0.5‰-classes. The maximum of 2 - 2.5‰ in $\Delta\delta^{18}O$ values was calculated only for two events, while for 23 of the 28 events $\Delta\delta^{18}O$ was 1.5‰ or less."*

Line 17: How was dryness or wetness of the canopy before an event started monitored? I assume that dry and wet is interfered from a certain period of time without rainfall. This should be described in the revised manuscript.

*Response: We inserted "…, i.e.after at least 6 hours without rainfall…" after "dry canopy"*

Line 23: If the information about Dd2H is important then I suggest  to take it out of the brackets and include it in the sentence with "and" and "respectively". Otherwise it should be omitted.

*Response: Changed as suggested.*

**Page 7**, Line 11-12: Is this in line with the literature? E.g. see (Qu et al., 2014). Please extend the discussion.

*Response: The discussion has been extended accordingly:*
*"Large intra-storm variabilities exist in the isotopic signature of $P_g$ and TF, which would have been impossible to detect when solely relying on commonly taken event-based bulk samples or even on data representing a higher sampling interval of typically 5 mm precipitation depth. We found that the variability of the continuous $P_g$ isotope data is higher than the variability of the continuous TF data (Fig. 6). For the latter, the signal is more dampened probably due to mixing processes, which was also found in other studies (Qu et al., 2013)."*

Line 21-22: Figure 3 only shows one significant correlation. Please correct this sentence.

Line 21: An explanation should be given why there is a negative correlation of rainfall intensity with interception loss. I can imagine that this is in line with what others have found for bulk samples. If possible respective references should be added.

Line 21-22: Is this in line with results reported in the literature? See e.g. (Dewalle and Swistock, 1994; Kato et al., 2013)

*Response: Figure 3 has been reworked and also covers more variables now. The respective paragraph has been rephrased to:*
*"In the data of the collected bulk samples a significant negative correlation between interception loss and the meteorological variable rainfall intensity as well as a moderate negative correlation between relative interception loss and depth of incident rainfall could be observed (Fig. 3).This means that the highest interception losses were found during events with either lowest rainfall intensities or with lowest depths. A weak positive relationship also existed between $\Delta\delta^{18}O$ and interception loss, indicating that $\Delta\delta^{18}O$ increases with increasing interception losses. This is in line with results found in other studies (Dewalle and Swistock, 1994; Brodersen et al.; 2000, Keim et al.; 2005, Kato et al., 2013; Allen et al., 2017). Only non-significant correlations were found between the other investigated quantities. …"*

Line 22: I don't understand what the authors mean with "There is no clear pattern for only one of these variables. . ."

*Response: We rephrased the respective sentence. It now reads:*

*"The small explained variance between any of the considered variables illustrates the complexity of the processes contributing to interception loss and the transformation of $P_g$ isotope ratios when becoming TF."*

Line 26-27: This needs to be related to what was reported earlier in the literature (e.g. Allen et al., 2013).

*Response: We rephrased the respective sentence to:*

*"This supports the interpretation that antecedent conditions have a clear impact on isotopic enrichment of TF which was also described in previous studies (Keim et al., 2005; Allen et al., 2014; Stockinger et al., 2015; Allen et al., 2017).*

There is more room for interpretation here. For instance, the difference of d18O in rainfall and throughfall is presented for different events and discussed, but the level of d18O of rainfall for different events is not presented.

*Response: We added the range of $δ^{18}O$ for the observed events to the results section. For the interpretation of interception processes we preferred to focus on the changes of the isotopic compositions ($Δδ^{18}O$) rather than the absolute numbers ($δ^{18}O$).*

For events for which the degree of rainout from Ocean to inland is low, levels of d18O of rainfall could be higher. Would it be possible that high initial levels d18O in rainfall lead to rather smaller increases of d18O from rainfall to throughfall?

*Response: We see your point but currently can't think of a physical reason that would support this hypothesis. We additionally looked at the relationship without finding any significant correlation.*

**Page 8**, Line 7-9: Scatterplots only presented for bulk samples.

*Response: We refer to the results shown in Fig. 4 to 6.*

Figure 1: For me it is not absolutely clear whether Pg is collected by the same collector for the water that goes through the tipping bucket and that one that is sampled. So is there only

one collector for each, Pg and TF, and the water go first through the tipping bucket and then the pump samples the water? Please clarify this in the figure. I think that the triangle within the box illustrates the tipping bucket, right? This must be shown more clearly.

*Response: We reworked Figure 1 as suggested to clarify this point.*

Figure 2: Add "depth" after "Time series of rainfall".

*Response: Changed as suggested.*

Figure 2: In the text it says that the event duration was only 2 hours, but in the legend a "three-hour bulk sample" is mentioned.

*Response: In order to also collect the water dripping from the canopy, the throughfall sampler was emptied later. We changed the legend to "bulk sample".*

Figure 2: Please provide the date of this event in the legend.

*Response: Added as suggested.*

Figure 3: I don't the point in part a) and b) of figure 3. The only additional information is the p-value. But the p-value should be added to the left side figure part for all correlations, e.g. in parentheses below the correlation coefficients. If the reason for a) and b) was to have interception on the x-axis, then it should be plotted as the first variable in the left side figure.

*Response: Figure 3 has been reworked and also covers more variables now. Detailed figures (a) an (b) have been removed. Levels of significance for all correlations have been integrated into the main figure as suggested.*

Figure 3: This is not a scatter plot of throughfall samples. Please rephrase the figure caption.
*Response: Changed as suggested.*

Figure 3: Please indicate in the figure that the intensity is that one of rainfall (and not throughfall).

*Response: Changed as suggested.*

Figure 3: Please explain in the legend what is shown the lower left und the upper right part of the figure on the left side. Include an explanation, why the size of the correlation coefficients differ. Is that really needed? The small numbers are hard to read.

*Response: In the reworked Figure 3 an explanation in the figure caption is added as suggested. The font size of the correlation coefficients has been unified.*

Figure 3: I suggest adding d18O of rainfall and throughfall and the length of the antecedent dry period to the scatter plot.

*Response: $\delta^{18}O$ data have been added to the scatter plot as suggested. We didn't show the data about the length of the antecedent dry periods because not all relevant time scales were sufficiently represented in our dataset.*

Figure 4: Please shift the right figure to the left side as it is mentioned first in the text.

*Response: Changed as suggested.*

Figure 5: Which of these events are shown in figures 2 and 6?

*Response: The event shown in Fig. 2 is not represented in Fig. 5. Information about which events are shown in Fig. 6 can be found in the new Table 1. A legend with the respective event IDs is added to Fig. 5.*

Figure 5: There are events for which Dd18O increases with time and others with an opposite trend. Does Dd18O correlate with d18O of rainfall per event?

*Response: We looked at this relationship without finding a significant correlation.*

Figure 6: Indicate the date of this event.

*Response: Changed as suggested.*

Figure 6: Delete "amounts," from the first line of the figure caption.

*Response: Changed as suggested.*

Figure 6: The colours of d_Pg and d_TF are hard to distinguish.

*Response: We changed the color to make the symbols more distinguishable.*

Figure 6: Why did throughfall start before rainfall?

*Response: Rainfall started before 19:27 already, but data are shown from the time when temperature at the contactor was stable.*

[revised manuscript text omitted]

---

## Author Response (AR2)

**Report #1**

*We thank the reviewer for the helpful comments, we changed the manuscript accordingly. Details about the changes are given below.*

*Please note: We inserted subsections in the **Discussion** section. In doing so, we re-arranged the paragraphs. The roman numbers at the beginning of each paragraph in the discussion section indicate the previous order (the roman numbers would be deleted prior to typesetting).*

I acknowledge that the authors have improved the manuscript, which now reads better than the previous version. I furthermore think that this is a valuable proof of concept of a novel method that finally could merit publication. Generally, I have the impression that the authors try to oversell their results with the help of claims that are not true. This devaluates the paper unnecessarily. I also recognize that several of my previous suggestions were not followed, which is of course always the right of the authors if they have convincing arguments to do so. Unfortunately, I am not convinced by their arguments concerning the following critics I have raised previously:

1. The title is misleading, because the presented method is not real time but has a delay of four minutes. Therefore, I repeat my suggestion to change the title to "Temporally highly resolved measurement of stable hydrogen and oxygen isotope ratios in water of rainfall and throughfall with a novel infrared laser-based method".

*We thank the reviewer for the detailed title suggestion.*

*However, we feel that the reviewer's title suggestion does not fully reflect the content we want to emphasize. Temporally highly resolved measurements had been feasible in the past already, yet by conventional analysis of vast amounts of discrete samples.*

*In order to find an acceptable compromise and according to the term "near real-time" that already appears in published literature in the context of similar approaches (Leis et al., 2011) we changed the title to "Continuous, near real-time observations of water stable isotope ratios during rainfall and throughfall events".*

2. The issue of the spatial representativeness of throughfall measurements, which frequently require a larger number of samplers, particularly if a high temporal resolution is envisaged, is not mentioned at all. The first conclusion stating that the proposed method is suitable for continuously observing the stable water isotope dynamics in precipitation and throughfall is not true, because the method in the current setting can only measure at a single, likely non-representative point, while all hydrological models require an area-representative measure. It is disappointing that the authors were not ready to simply discuss this restriction, which is another attempt to oversell their results.

*There seems to be a misunderstanding. Again, we agree that e.g. for catchment-scale investigations isotope data need to reliably cover the spatial variability of gross precipitation and throughfall. The aim of this study, however, was to provide a proof of concept of a novel method that allows replacing the commonly temporally poorly resolved analysis of e.g. event-based point-level bulk samples. Therefore, throughout the manuscript we emphasize the temporal resolution, not the spatial representativeness. Starting in the objectives, we aim at point-level observation (p3, l15). Also in the discussion we explain, that the spatial scale and variability needs to be considered in future studies (p10, l6f). And finally, in the conclusion we state that the proposed method needs to be employed across larger spatial scales in order to contribute to the aim of thorough isotopic sampling of throughfall (p11, l16).*

*To further emphasize the issue of spatial representativeness we added*

*"For the sampling points in our study,"*

*to the following sentence at the beginning of the discussion*

*"We found that the variability of the continuous $P_g$ isotope data is was higher than the variability of the continuous TF data (Fig. 6)."*

*Additionally, we expanded the discussion with the following paragraph:*

*"On the other hand, larger or multiple collectors would ensure a better representation of the spatial variability of Pg and TF isotope data. This issue is critical for the thorough investigation of spatially heterogeneous processes affecting isotope input data at e.g. the catchment scale. A better representation of spatially distributed isotope data based on the proposed method can be achieved by installing a representative number of collectors from where water is continuously aggregated and directed to one analyzer. This approach would consider the spatial heterogeneity similar to the roving collector approach (Allen et al., 2015) but at the same time it still does not allow for the quantification thereof. For the latter purpose, the entire setup described in this study could be multiplied in order to analyze individually the continuous water samples from the respective number of collectors considered necessary for covering the spatial heterogeneity. However, given the required number of suitable isotope analyzers this approach may not be feasible for many research groups."*

3. The authors show in Fig. 6 that the "deuterium excess" value changes with time in both rainfall and throughfall. This finding is just mentioned but not discussed. What do we learn from this change? You should at least present a speculation or not show the result. The cited reference of Gat (1996) suggests e.g., that a change of the deuterium excess is related with different water sources or with a changing contribution of re-evaporated water (from isotopically distinct water pools). Which role do (bio)chemical reactions play? Mass-dependent kinetic fractionation of water during phase transition – which is generally non-equilibrium! - will always change the isotope ratios of both, H and O in a related way. To explain a changing deuterium excess, you need a specific deuterium (or 1H) source (free of 18O), e.g., from organic and mineral acids.

*We extended the discussion and replaced*

*"The additionally calculated Δd values did not show a clear or systematic pattern that would add to previous findings."*

*by*

*"Similar but less distinct clusters could be observed in the derived Δd values. The time series of Δd from initially dry canopies were quite stable and predominantly negative. In contrast, we observed higher, mainly positive and more fluctuating Δd values from initially wet canopies (Fig. 5 (c)). The fluctuations may be attributed to changing meteorological conditions affecting evaporation but also to the fact that Δd values are quantities derived from four isotope measurements ($\delta^{18}O$ and $\delta^{2}H$ of both $P_g$ and TF) and therefore bear generally higher uncertainties due to error propagation. The positive Δd values are not consistent with our expectation that evaporation would persistently lead to negative Δd values and it stands out that the highest Δd values were calculated from continuous samples with highest evaporative enrichment in heavy isotopes of both $\delta^{18}O$ and $\delta^{2}H$ (Fig. 5). Regardless of the sampling method, we found positive Δd values in both continuous (Fig. 5 and Fig. 6 (qualitatively)) and bulk sample data (Fig. 3). This indicates that the appearance of positive Δd values was not a methodological artefact due to our way of continuous data interpretation but had to be physically based. Mathematically, positive Δd values would have resulted from evaporation lines with*

*slopes higher than the meteoric water line. However, we are not aware of such a phenomenon.*
*Positive Δd values as a result of mixing processes, as suggested elsewhere (Allen et al., 2017), would*
*still necessitate at least one substantial endmember with significantly higher than evaporation-only-*
*caused d values. The hypothetical formation of such endmembers remains unclear and was not*
*indicated by the continuous $P_g$ data."*

Moreover, I still think that the "double delta" notation used by the authors to express differences
between two delta values is highly unusual and that Figs. 2 and 6 should be combined.

*We think that it is mathematically correct and also provided a definition (Eq. (3) to (5)).*
*Further, the "double delta" notation is consistent with published literature (see e.g. Allen et al., 2014,*
*2015, 2017; Stockinger et al., 2015) where these articles were published in four different journals. We*
*therefore prefer to keep the "double delta" notation for differences in delta values.*

*Figures 2 and 6 represent different datasets from different events and they highlight different aspects*
*of our method. This is detailed in the respective parts of the discussion; therefore we prefer to keep*
*them separated.*

Further to these major concerns, I again offer a number of more minor suggestions to improve the
manuscript.

p. 1, l. 3: water isotopic composition

*We assume that l. 8 was meant where we now added "water" before "isotopic composition".*

p. 1, l. 15: Delete "and thereby,".

*Changed as suggested.*

p. 1, l. 18: This comparison might be possible with respect to the temporal resolution but not to the
spatial representativeness.

*In order to address this issue, we rephrased the following sentence to*
*"Future improvements of the spatial representativeness will make our approach an even more*
*powerful tool towards detailed insight in the dynamic processes contributing to interception during*
*rainfall events."*

p. 1, l. 23: Define for what the tracer is ideal. Usually, the stable isotope ratios of H and O are far
from ideal, because differences among various sources are frequently (too) small and the signal is
prone to changes as a consequence of kinetic isotope fractionation or mixing and thus not stable.

*We extended our argument by adding*
*"Thus, any change of the stable isotope ratios reflects the conditions a given water reservoir has been*
*exposed to prior to sampling. This makes stable isotopes superior to e.g. solute concentrations, which*
*may be prone to precipitation, degradation or chemical reactions."*

p. 1, l. 25-26: This is mostly true for the long-term mean isotopic signature of rainfall water (>1 yr)
but hardly if short time periods are considered as is done in this study.

*This sentence is meant to be an introductory sentence.*

p. 2, l. 3: "amount" was changed to "depth" but not consistently. Please also replace the remaining "amount".

*Changed as suggested.*

p. 2, l. 11-13: Who does this? Seems trivial.

*We agree. Here we refer to the statements of the cited literature.*

p. 2, l. 7-8: I did not understand this sentence.

*We rephrased the sentence to*
*"These effects result in offsets of TF, they reduce the depth of TF compared to $P_g$ and cause the difference in $\delta^{18}O$ between event-based sampled TF and $P_g$."*

p. 2, l. 9-11: What is the volume-weighted mean of the relative interception loss? Usually, the cumulative interception loss in percent of rainfall during a defined period is given.

*We agree. Here we refer to the statements of the cited literature.*

p. 2, l. 17-19: The patterns of throughfall are indeed heterogeneous but I didn't understand what you mean by "when related to leaf area index as well as to spatial variability in general".

*We changed this sentence to*
*"The study showed that the spatial patterns of TF were very heterogeneous and ecosystem dependent when related to leaf area index (LAI) or other biotic factors (Levia et al., 2011)."*

p. 2, l. 22-24: The method of roving throughfall collectors is used to improve the representativeness of the throughfall measurement with a limited number of samplers. How could it help to measure the size of spatial variability? The latter is usually reached with the help of a sufficiently high number of samplers, which are not roved.

*We replaced "observation by "representation".*

p. 3, l. 17: Replace two times "amount".

*Changed as suggested.*

p. 4, l. 19: Unclear. Did you collect volume-proportional subsamples, which you combined to a bulk sample? I know "volume-weighted" mainly in the context of solute concentrations. Perhaps volume-proportional is the better term at all places, where you use "volume-weighted"?

*Thank you for alerting us to the misleading wording. We rephrased the sentence to*
*"The overflow and the excess water downstream of the membrane module were collected and analysed separately. From these isotope results volume-weighted means were calculated to represent*

*bulk sample values for each continuously measured event. Additional bulk-samples were physically collected and analysed for events when low intensities or the occurrence during night-time prevented continuous analysis."*

p. 4, l. 26: leaves

> *Changed as suggested.*

p. 4, l. 28: 15 m is a large distance in a forest. There are possibly several other trees in between.

> *This is the distance between the point of meteorological observations and the $P_g$ collector. The distance to the TF collector was much smaller with no other tree in between.*

p. 6, l. 6-7: This can be seen in the figure and should therefore usually not be repeated in the text.

> *The sentence has been removed. We presented the contained information earlier, in the context of the continuous data.*

p. 6, l. 15 (and throughout the manuscript): I suggest to consistently report results in the past tense.

> *Changed as suggested.*

p. 7, l. 18-p. 8, l. 2: Switch to past tense.

> *Changed as suggested.*

p. 8, l. 16-18: I did not understand this statement. What did you expect and why?

> *We rephrased the sentence to*
> *"However, this is not reflected in continuous $d_{TF}$ values (Fig. 6). We would have expected that they follow the trend of $d_{Pg}$ while being comparatively lower as a result of non-equilibrium fractionation. Neither was the case."*

p. 8, l. 29: detectable

> *Changed as suggested.*

p. 9, l. 16: Who prefers "large and long surfaces"? I would prefer small and many separate surfaces. Large and long surfaces get quickly too dirty in forests so that their use is just not feasible in the practice.

> *We discussed this aspect in detail by adding the following paragraph*
> *"On the other hand, larger or multiple collectors would ensure a better representation of the spatial variability of Pg and TF isotope data. This issue is critical for the thorough investigation of spatially heterogeneous processes affecting isotope input data at e.g. the catchment scale. A better representation of spatially distributed isotope data based on the proposed method can be achieved by installing a representative number of smaller collectors from where water is continuously*

*aggregated and directed to one analyzer. This approach would consider the spatial heterogeneity similar to the roving collector approach (Allen et al., 2015) but at the same time it still does not allow for the quantification thereof. For the latter purpose, the entire setup described in this study could be multiplied in order to analyze individually the continuous water samples from the respective number of collectors considered necessary for covering the spatial heterogeneity. However, given the required number of suitable isotope analyzers this approach may not be feasible for many research groups."*

p. 10, l. 6: Why should a mere flow have an isotope effect? In fact, it does not!

*We changed the sentence to*
*"Horizontal water redistribution in the canopy via flow along branches may result in mixing of different water reservoirs. This may have an effect on the observed isotopic composition as it also changes the unknown spatial pattern of precipitation water isotopes."*

Fig. 2: Add the abbreviation "vapour_mov" after "moving average" in the legend. From where does the temperature originate? From the station at 15 or at 250 m from the tree?

*Changed in the legend as suggested.*
*We added "in 250 m distance" after "air temperature".*

Fig. 3: I already recommended not to show regression lines if there is no significant correlation, because they are misleading. You should explain all abbreviations used in the figure in the legend.

*Changed as suggested.*

Fig. 6: The term "vapour sampling" is likely misleading, because I assume that you converted the vapour measurement to liquid values. Otherwise the measurements of the discrete liquid samples would likely not match so well. Where were temperature and vapour pressure deficit measured (15 vs. 250 m away from the tree)? Anyway, I think that this figure is not needed.

*We inserted "liquid water" after "Time series of"*

*Temperature and relative humidity data were recorded in 15 m distance from which vapour pressure deficit was calculated.*

*This figure reveals several aspects especially regarding the contrast between $P_g$ and TF that cannot be seen in e.g. Fig. 2. We therefore prefer to keep it.*

*Cited literature:*

Leis, A., Plieschnegger, M., Harum, T., Stadler, H., Schmitt, R., Pelt, A.v., and Zerobin, W.: Isotope Investigations at an Alpine Karst Aquifer by Means of On-Site Measurements with High Time Resolution and Near Real-Time Data Availability, in: International Symposium in Isotopes in Hydrology, Marine Ecosystems and Climate Change Studies, 2011.

**Report #2** (Referee #3)

*We thank the reviewer for the helpful comments, we changed the manuscript accordingly. Details about changes are given below.*

*Please note: We inserted subsections in the **Discussion** section. In doing so, we re-arranged the paragraphs. The roman numbers at the beginning of each paragraph in the discussion section indicate the previous order (the roman numbers would be deleted prior to typesetting).*

This is an interesting approach for the collection of high resolution rainfall and throughfall data on isotopic composition.

*Thank you!*

All my comments have been addressed. The paper has been improved compared to the first version and merits publication after correcting a few minor aspects:
Page 1, Line 27: Delete one of the two points.

*Changed as suggested.*

Page 2, Line 22: Shift the reference to Line 20.

*Changed as suggested.*

Page 6, Lines 11-13: Adapt variables according to your changed figure 2.

*We assume that Fig. 3 was meant, where we adapted the variables in the figure to make it consistent with the text.*

Page 6, Line 15-16 and page 9, Line 4-5: The mentioned "weak positive relationship" is not significant and, hence, there is no relationship.

*We deleted the respective sentences.*

Page 6, Line 19 and Page 9, Line6-7: Yes there are two other significant correlations that are not mentioned in the text. Please correct.

*Thank you for the hint. In the results section we inserted:*

*"Also a weak negative correlation between interception loss and rainfall depth was found, as well as a weak positive correlation between the logarithm of the mean rainfall intensity and rainfall depth."*

*before "The isotopic composition of $P_g$ ranged from…"*

*In the discussion we reworked the respective paragraph, it now reads:*

*In the data of the collected bulk samples the isotopic composition of $P_g$ and TF were highly correlated to each other. This concurs with our expectation, as does the quasi persistent enrichment in heavy isotopes of TF relative to $P_g$ (positive $\Delta\delta^{18}O$ values) indicating evaporation from the canopy. Although expected, no significant positive correlation between $\Delta\delta^{18}O$ and relative interception loss was found.*

*The same applies for potential negative correlations between Δd and Δδ¹⁸O or relative interception loss, which were also not found. However a moderate negative correlation between interception loss and the logarithm of the meteorological variable rainfall intensity, a weak negative correlation between relative interception loss and depth of incident rainfall as well as a weak positive correlation between the logarithm of the mean rainfall intensity and rainfall depth could be observed (Fig. 3). This means that the highest interception losses were found during events with either the lowest rainfall intensities or with the lowest depths and that higher rainfall intensities result in higher total rainfall depths. This is in line with results found in other studies (Dewalle and Swistock, 1994; Brodersen et al.; 2000, Keim et al.; 2005, Kato et al., 2013; Allen et al., 2017). Only non-significant correlations were found among the other investigated quantities. The generally small explained variance among any of the considered variables illustrates the complexity of the processes contributing to interception loss and the transformation of $P_g$ isotope ratios when becoming TF.*

Page 6, Line 23: Close the parentheses before the point.

> *Changed as suggested.*

Table 1: Please indicate that this is the mean intensity as it could also be e.g. the maximum intensity per event. For event #3 the parentheses are also needed for throughfall mean intensity.

> *Both changes done as suggested.*

Page 8, Line 1 and Figure 6: Use the same labels in the figure and in the text as well as in the figure caption as dTF and d_TF is used right now.

> *We changed the labels in the figure.*

Page 9, Line 3: Add a space after the point.

> *Changed as suggested.*

Page 10, Line 29-30: This has been shown by whom? Please add a reference.

> *Reference added as suggested.*

Page 10, Line 34: A reference is missing after "water".

> *We assume that line 33 was meant where a reference has now been added.*

**Report #3** (Referee #2)

*We thank the reviewer for the helpful comments, we changed the manuscript accordingly. Details about changes are given below.*

**General comments**:

I thank the authors for their efforts to improve the manuscript following the suggestions of the reviewers.

The authors satisfactorily answered many of my specific comments and made the corresponding changes in the manuscript. Yet, some answers to my general comments have been omitted and a few of my specific comments remained unanswered, which I thought were among the most interesting aspects to be discussed.

Moreover I think that the manuscript would still benefit from a clearer presentation of the ideas currently presented in the introduction and in the discussion. More context about the hydrological applications of stable isotopes (e.g. hydrograph separation, end-member mixing analysis, and transit time calculations) would improve the introduction and make the manuscript and its importance more compelling. The discussion also still needs a clearer explanation to why the presented advances in isotope measurement techniques are important for such applications and/or for stimulating the creation of new applications currently challenged by limited isotopic data. Right now the manuscript stops right before making the step. The readers need to be completely convinced that it is not enough to simply take bulk samples in TF and in Pg.

Therefore, I recommend an additional round of revisions. I believe this work will be very useful, yet it contains only little interpretation of the data in terms of interception processes and the added value of the approach is not convincingly presented. Everything is there, but the authors cannot leave this to the reader.

*Following the suggestions of the reviewer we re-arranged and expanded the introduction regarding isotope applications in hydrology in general as well as in interception studies and added further references. Furthermore, we substantially reworked and restructured the discussion section thereby adding several paragraphs that focus on process related aspects.*

**Specific comments**:

L6: Please add "with the water vapor" after "diffusive exchange" to make it more specific.

*Changed as suggested.*

L15: Please add "on the stable isotopes of water" after "the temporal effects of interception processes" to make it more specific.

*Changed as suggested.*

L23: One could add "from the use of stable isotopes of water" after "events" to make it clearer that the advances presented in the approach are about isotope measurement techniques.

*We inserted "of water stable isotope data" after "comparison" in the previous sentence.*

Introduction: Until now, it is still difficult to follow the argumentation in the introduction. I think all ideas are there, but they are not well organized. For me, the introduction should be organized along these key points (just a suggestion):

• The role of stable isotopes as the input for many hydrological applications (with more references to these specific applications) and as a tool to better understand interception processes (refs)
• The often neglected differences between Pg and TF amounts and isotopic composition (refs).
• The consequences of this neglected difference for the mentioned applications in isotope hydrology (refs + personal interpretation).
• The possible interception mechanisms causing these differences and the current research need to better understand these mechanisms (refs).
• Despite recent progress in measurement techniques (refs), the current lack of high-resolution isotopes in both TF and Pg to correct the estimation of the true isotopic inputs and to shed light on interception mechanisms.
• The obvious goal of the paper.

*The introduction has been re-arranged as suggested and significantly expanded (additional refs were inserted) following your suggestions.*

L25: You changed "signatures" to "ratios" but my point was that nothing should follow "stable isotopes" to be exact. For example you would not say that chloride concentration is a tracer. Chloride itself is a tracer.

*Sorry for the misunderstanding, we deleted "ratios".*

L26: adding "paths" after "flow" would make more sense.

*Changed as suggested.*

L4-6: This sentence is still not clear, and it is still too long. Don't use so many commas. This is not the only long sentence that needs to be restructured.

*We rephrased this sentence (among others) to*
*"They are driven by evaporation from the canopy during or between storms as well as by diffusive exchange with ambient vapour. Furthermore, redistribution in the canopy and storage effects where water is differentially retained or mixed contribute to interception."*

L25-27: This sentence is not clear, please reformulate it. "spatial variability in general" was not clarified as requested, please change it.

*The sentence has been changed to*
*"The study showed that the spatial patterns of TF were very heterogeneous and ecosystem dependent when related to leaf area index (LAI) or other biotic factors (Levia et al., 2011).*

Line 15-16: That was not what I meant. I meant that the instrument setup could induce a moving average with time window up to 36 s long applied to the isotopes being pumped to the membrane modules. This smoothing happens before the 90 s moving average is applied to the isotope readings. This 36 s moving average is the result of the physical configuration of the setup because some water could accumulate in the smaller 3 mL funnel during rainfall intensities larger than 5 mL/min. Of course the 90 sec moving average discards the variability of the data happening during periods shorter than 90 sec. But from the measurement protocol, it seems that during rainfall events with

large intensities (i.e. when Vd is full with water for some time), the isotope readings every 4 sec will actually correspond to a moving average of 36 sec. Please mention it in the discussion.

*We referred to this aspect in the first paragraph of the discussion, where it now reads: "Given the dead volume of the perfused components the collected isotope data were subject to averaging with a time window of ~36 s. Additionally, we calculated a moving average with 90 s integration time for data noise reduction. Nonetheless, large intra-storm variabilities…"*

L28: I know one reviewer asked to change "compared" to "correlated", but it now sounds like there is a correlation between all the variables in the plot. Why not change it to "variables […] are plotted against each other to check for correlations between them"?

*Changed as suggested.*

Discussion: Why not use subsections in the discussion? That would make it clearer.

*We inserted subsections as suggested. In doing so, we re-arranged the paragraphs. The roman numbers at the beginning of each paragraph in the discussion section indicate the previous order (the roman numbers would be deleted prior to typesetting).*

L6: "mixing processes": where?

*We added "in the canopy"*

L7: "indicates" is better than "indicative for"

*Changed as suggested.*

L14: Why did this last another two days? Do you mean that the analysis was completed only two days after the samples were brought back to the lab?

*We meant that assuming sufficient lab capacities, the net measurement time in the lab for this amount of samples was two days.*

L23-27: These two sentences are a repetition of what is written in the results section. Either rewrite, shorten and add interpretations, or remove. These sentences would actually fit much better in the conclusion which is missing these relevant results!

*We rephrased the section to:*

*"In the data of the collected bulk samples the isotopic composition of $P_g$ and TF were highly correlated to each other. This concurs with our expectation, as does the quasi persistent enrichment in heavy isotopes of TF relative to $P_g$ (positive $\Delta\delta^{18}O$ values) indicating evaporation from the canopy. Although expected, no significant positive correlation between $\Delta\delta^{18}O$ and relative interception loss was found. The same applies for potential negative correlations between $\Delta d$ and $\Delta\delta^{18}O$ or relative interception loss, which were also not found. However a moderate negative correlation between interception loss*

*and the logarithm of the meteorological variable rainfall intensity, a weak negative correlation between relative interception loss and depth of incident rainfall as well as a weak positive correlation between the logarithm of the mean rainfall intensity and rainfall depth could be observed (Fig. 3). This means that the highest interception losses were found during events with either the lowest rainfall intensities or with the lowest depths and that higher rainfall intensities result in higher total rainfall depths. This is in line with results found in other studies (Dewalle and Swistock, 1994; Brodersen et al.; 2000, Keim et al.; 2005, Kato et al., 2013; Allen et al., 2017). Only non-significant correlations were found among the other investigated quantities. The generally small explained variance among any of the considered variables illustrates the complexity of the processes contributing to interception loss and the transformation of $P_g$ isotope ratios when becoming TF."*

Conclusion: A sentence or two about the key results of the study as from the measurements is missing.

*We replaced "The lack of strong correlations between the investigated rainfall characteristics illustrates the complexity of interception processes."*

*by*

*"Significant correlations of $P_g$ and TF depths and depth-derived quantities were found as expected and concurred with findings from other studies. The lack of significant correlations involving isotope derived quantities cannot be explained with our current knowledge and process understanding and calls for further scrutiny. Especially the high abundance of positive $\Delta d$ values should be subject of future studies."*

L7: Please add "and stable isotope measurements" after "characteristics". You looked at the stable isotopes in rainfall as well.

*Changed as suggested.*

Figure 3: Please make the symbol associated with p<0.05 larger. Why is there a symbol for p<0.01? It is not used. Please add "of precipitation and throughfall" after "bulk samples" in the caption.

*We changed the symbols as suggested.*

*We now listed all parameters that are compared in the scatter plot matrix in the caption.*

Figure 5: "deviations" is not consistent with previous occurrences of "differences" for this metric. I think you should use only "differences" throughout the manuscript. Please make sure this is done consistently.

*Changed as suggested.*

***Additional literature:***

[revised manuscript text omitted]

Level of significance:   ·  p < 0.05      *  p < 0.01      **  p < 0.001      ***  p < 0.0001

[Figure]

**Figure 3: Scatter plot matrix of relative interception loss, log of mean $P_g$ intensity, $P_g$ depths and isotope-related characteristics ($\Delta\delta^{18}O$, $\Delta d$, $\delta^{18}O_{Pg}$ and $\delta^{18}O_{TF}$) derived from 28 event-based bulk samples. Upper right part: Pearson correlation coefficients and level of significance (stars); lower left part: scatter plots and linear regressions (red lines).**

[Figure]

**Figure 4: Difference of the isotopic signature ($\Delta\delta^{18}O$) between TF and $P_g$ for (a) 28 event-based bulk samples and (b) 9 continuously analysed events of the same period.**

[Figure]

Figure 5: Time series of differences between the isotopic signature of TF and P$_g$, (a) $\Delta\delta^{18}$O, (b) $\Delta\delta^2$H, (c) $\Delta d$; events on initially dry canopy (dashed line) and on wet canopy (solid lines).

[Figure]

[Figure]

**Figure 6: Time series of liquid water $\delta^{18}$O and d-excess (_d_) derived from vapour sampling of P$_g$ (blue) and TF (green), discrete liquid samples (big circles), air temperature (T$_a$) (small blue circles) and vapour pressure deficit (VPD) (red) of the rainfall event from August 4th 2016. Time series of $d_{Pg}$ (dark blue dots), $d_{TF}$ (brown dots) and T$_a$ are referenced on the right vertical axes. Periods of intensities below threshold for the continuous sampling method (bubbles at membrane contactor) are shown in light blue (P$_g$) and light green (TF).**

---

## Author Response (AR3)

*We thank the reviewer for reviewing the manuscript again and his helpful comments. We changed the manuscript accordingly.*
*Details about the changes are given below.*

*Line numbers refer to the plain text version of the revised manuscript.*

**Report #1** (Referee #2)

I thank the authors for their efforts to improve the manuscript. I am satisfied with the changes to the manuscript that tackled my suggestions/demands. Changing the order of the ideas in the discussion would eventually give the work a better readability and impact. I also have a few more minor comments. These changes can probably be made without an additional round of revisions.
L26: "superior" may be too strong and start a debate with geochemists. Please consider another adjective. Also note that while isotopes are not affected by chemical reactions like solutes, stable isotopes nevertheless go through fractionation processes. These make it hard to link isotopic ratios to other processes (e.g. flows) in the absence of controlled conditions (e.g. temperature and humidity).

> *We changed "superior" to "advantageous".*

L6: De Walle is not an appropriate citation, cite rather Liu et al. (2004), https://doi.org/10.1029/2004WR003076.

> *Changed as suggested.*

L7: I think it is "von Freyberg" et al.

> *Changed as suggested.*

L14: "catchment residence times"

> *Changed as suggested.*

L15-16: This sentence is really important. Splitting it and adding more details would give it more impact.

> *We splitted and rephrased the sentence. It now reads:*
>
> *"Many studies have used the temporal dynamics in the isotopic composition of precipitation for estimating catchment residence times. Particularly, on forested sites where meteorological and isotopic reference stations are generally in the open interception losses and accompanying isotope effects must be considered as they have a significant impact on the input function (…)."*

L27: I would replace "precipitation input" with "the water effectively recharging the catchment"

> *Changed as suggested.*

L30: "decreasing TTDs" does not make much sense. Write "lower travel times" instead.

*Changed as suggested.*

L32: "data" -> "isotopes"
Discussion:
The discussion has almost all the ideas needed to give the work enough impact. However, these ideas are presented in a way that makes it challenging to follow and makes the reading a bit painful. I recommend that the authors really focus on their technological progress offering the benefits of continuous measurements against bulk samples at the start of the discussion. Then, they should move on to interpretations of their correlations and plots in terms of processes and applications to models/isotope hydrology applications, and then only technical aspects + limitations/way forward. Therefore, I suggest the following order (using the current section numbers) and advice for improvement:
(1) 4.1: Another title would describe the contents better.

*Changed to "4.1 Continuous measurements".*

Paragraph II seems to fit better in 4.3.

*Changed as suggested.*

L25: You could mention that this finding about dTF too is possible only with continuous data and that further investigations about this will need such a measurement setup.

*In section 4.7 we added*

*"For validation purposes, it would then require high resolution meteorological and isotope data as available from our setup in order to match the resolution of the envisioned modeling time steps."*

*Further, in the Conclusion section we added*

*"The obtained data will be crucial for mechanistic modelling approaches which will yield more realistic isotope input functions and thereby improve water flow and solute transport estimations for vegetated catchments."*

L31: You could mention that this also dispenses the transport/storage of many samples.

*In section 4.1 we inserted "Additionally, continuous measurements dispense the transport and storage of large quantities of samples."*

*and also referred to it in the Conclusion section (p. 13, l.29) by inserting*

*"..., transporting and storing"*

Also, avoid using "obviously" (found also in other sentences). What is obvious for someone may not be for someone else.

*We rephrased to: "Generally, there was a tremendous loss of…".*

(2) 4.6: the sentences lines 9-14 and the first sentence of paragraph IX seem to fit better in 4.5.

*Changed as suggested.*

Paragraph X would be better directly after VI.

*Changed as suggested.*

(3) 4.7: Make it clear here that only the continuous measurements in isotopes can validate such a mechanistic model. A word about the isotope applications in hydrology to echo the contents of the intro?

*We added to section 4.7:*

*(l. 9) "…as well as intra- and inter-storm variabilities…" and*

*(l. 22) "Such a model would derive from $P_g$ data a more realistic isotope input function of water effectively recharging forested catchments. For validation purposes, it would then require high resolution meteorological and isotope data as available from our setup in order to match the resolution of the envisioned modeling time steps."*

*To the Discussion section we added:*

*"The obtained data will be crucial for mechanistic modelling approaches which will yield more realistic isotope input functions and thereby improve water flow and solute transport estimations for vegetated catchments."*

L14: Not clear what simulation is meant.

*We inserted "envisioned" before "simulation".*

"needed" -> "needs"?

*Changed as suggested.*

L14-16: Not clear, break down / reformulate.

*We inserted "envisioned" before "simulation time step".*

(4) 4.2
L15: "meteorological variable" seems unnecessary

*Deleted as suggested.*

(5) 4.4
L5: Similar but less distinct than what?

*We rephrased the sentence to*

*"In the derived Δd values two clusters could be observed that were similar to but less distinct than the difference in $\delta^{18}O$ and $\delta^2H$ of the continuous measurements."*

L8: But why would these uncertainties be higher for initially wet conditions?

*We inserted*

*"In the case of initially wet canopies, mixing with pre-event water which was inconsistently subjected to evaporative enrichment of heavy isotopes may have contributed to the observed higher ⬚d variabilities."*

L14-15: Really? For me this is not obvious. This needs further explanation (show the equations allowing this conclusion?)

*We rephrased the sentence to*

*"Conceptually, positive Δd values could have resulted from evaporation lines with slopes higher than that of the meteoric water line, causing TF isotope values to plot above the meteoric water line in dual isotope space."*

(6) 4.3
L24: The first sentence is repetitive w.r.t. the last sentence of 4.2. This may be irrelevant after rearranging the order of the paragraphs.

*Yes, this is now irrelevant.*

L27: Alterations in the isotopic signal evidenced by interception losses? That does not make sense. Please explain further.

*We rephrased the sentence to*

*"Typically, air temperature as well as vapour pressure deficit being the main driver of evaporation slightly decreased over the course of an event. However, evaporation as evidenced by the observed interception losses still occurred and altered the isotopic signal."*

(7) 4.5
Conclusion: Currently this section lacks enough detail about the impact/consequences for isotope hydrology. It is nice approach, but how will it impact the field. Does it really matter?
Page 14:
L8-10: More details about this and applications in isotope hydrology.

*We rearranged the Conclusion section and inserted*

*"The obtained data will be crucial for mechanistic modelling approaches which will yield more realistic isotope input functions and thereby improve water flow and solute transport estimations for vegetated catchments."*